# A Unified Sampling Framework for Solver Searching of Diffusion Probabilistic Models

**Enshu Liu**
Department of EE, Tsinghua University &
Infinigence-AI
les23@mails.tsinghua.edu.cn

**Xuefei Ning**[*]**, Huazhong Yang, Yu Wang**[*]
Department of EE, Tsinghua University
foxdoraame@gmail.com
yanghz@tsinghua.edu.cn
yu-wang@mail.tsinghua.edu.cn

## Abstract

Recent years have witnessed the rapid progress and broad application of diffusion probabilistic models (DPMs). Sampling from DPMs can be viewed as solving an ordinary differential equation (ODE). Despite the promising performance, the generation of DPMs usually consumes much time due to the large number of function evaluations (NFE). Though recent works have accelerated the sampling to around 20 steps with high-order solvers, the sample quality with less than 10 NFE can still be improved. In this paper, we propose a unified sampling framework (USF) to study the optional strategies for solver. Under this framework, we further reveal that taking different solving strategies at different timesteps may help further decrease the truncation error, and a carefully designed *solver schedule* has the potential to improve the sample quality by a large margin. Therefore, we propose a new sampling framework based on the exponential integral formulation that allows free choices of solver strategy at each step and design specific decisions for the framework. Moreover, we propose $S^3$, a predictor-based search method that automatically optimizes the solver schedule to get a better time-quality trade-off of sampling. We demonstrate that $S^3$ can find outstanding solver schedules which outperform the state-of-the-art sampling methods on CIFAR-10, CelebA, ImageNet, and LSUN-Bedroom datasets. Specifically, we achieve 2.69 FID with 10 NFE and 6.86 FID with 5 NFE on CIFAR-10 dataset, outperforming the SOTA method significantly. We further apply $S^3$ to Stable-Diffusion model and get an acceleration ratio of $2\times$, showing the feasibility of sampling in very few steps without retraining the neural network. Our code is available at https://github.com/jsttlgdkycy/USF.

## 1 Introduction

Diffusion probabilistic models (DPMs) (Sohl-Dickstein et al., 2015; Ho et al., 2020; Song et al., 2020b) have emerged as a new generative modeling paradigm in recent years. DPMs model the probability distribution through training a neural network to estimate the score function or other equivalent form along pre-defined forward diffusion stochastic differential equations (SDEs) (Song et al., 2020b). Sampling from DPMs can be viewed as solving the corresponding reverse diffusion SDEs (Song et al., 2020b; Ho et al., 2020) or the diffusion ODEs (Song et al., 2020b;a) by discretizing the continuous timesteps.

Despite the high generation ability, the main drawback of DPMs is the slow sampling speed due to the large number of discretized timesteps for the numerical accuracy of the DE solver (Song et al., 2020b; Ho et al., 2020) with neural inference on each step. Therefore, accelerating the sampling process of DPMs is very meaningful.

Current work focusing on decreasing the DPM sampling steps can be divided into two categories. The first of them needs retraining of the neural network (Luhman & Luhman, 2021; Salimans & Ho, 2022; Meng et al., 2023; Song et al., 2023), which takes significant extra costs, especially for large models like Rombach et al. (2022). The other branch of works attempts to design efficient solvers of differential equations (DEs) to accelerate DPMs without retraining the existing off-the-shelf models (Song et al., 2020a; Watson et al., 2021; Bao et al., 2022; Liu et al., 2022; Lu et al., 2022a;b;

---

[*]Corresponding Authors

Zhang & Chen, 2022; Zhang et al., 2022; Zhao et al., 2023). State-of-the-art methods utilize the equivalent exponential integral form of the reverse diffusion ODE, accurately calculating the linear term while estimating the integral term through various high-order numerical methods (Zhang & Chen, 2022; Lu et al., 2022a;b; Zhao et al., 2023). These methods can reduce the number of steps to 15∼20 steps, but the performance decreases rapidly with lower than 10 steps.

We notice that many solving strategies of these solvers are empirically set and kept constant along the sampling process except for a few special timesteps, leading to suboptimal performance with inadequate timesteps. In this paper, to systematically study the impact of these strategies, we propose a unified sampling framework based on the exponential integral form of the diffusion ODE, named USF, which splits the solving process of one step into independent decisions of several components, including the choice of 1. *timestep*, 2. *starting point of the current step*, 3. *prediction type of the neural network*, 4. *order of Taylor expansion*, 5. *derivative estimation method*, and 6. *usage of ODE correctors*. Based on this framework, we reveal that the quality and efficiency of training-free samplers can be further improved by designing appropriate *solver schedules*, motivated by the key observation that the suitable solving strategies vary among different timesteps. Therefore, *solver schedules*, which means the different solving strategies assigned to each timestep, is very important for the sample quality. Our proposed framework can incorporate the existing diffusion solvers by assigning corresponding decisions to those components and allows the ensemble of different solving strategies in the timestep dimension, enabling sufficient potential to outperform existing sampling methods. In addition, we also design new strategies different from existing diffusion solvers for the derivative estimation component, making the proposed sampling framework more promising.

However, designing solver schedules is difficult due to the vast decision space. To address this problem, we propose $S^3$ to search for optimal solver schedules automatically. We construct a performance predictor to enable the fast evaluation of one solver schedule and use it to guide a multistage search process to find well-performing solver schedules under a certain budget of number of function evaluation (NFE). Our contributions are summarized as follows:

- We propose a new sampling framework for DPMs, called USF, which unifies existing diffusion ODE solvers based on exponential integral, to systematically and conveniently study the strategies chosen for diffusion samplers. Based on this framework, we design some new solver strategies, including using different solver strategies across timesteps, low-order estimation for derivatives, more types of scaling methods and searchable timesteps.
- We propose a predictor-based multi-stage search algorithm, $S^3$, to search for the well-performing solver schedule under a certain NFE budget. Our method can directly utilize off-the-shelf DPMs without any retraining of the diffusion neural network and find outstanding solver schedules with a moderate search cost, demonstrating its practical applicability.
- We experimentally validate our method on plenty of unconditional datasets, including CIFAR-10 (Krizhevsky et al., 2009), CelebA (Liu et al., 2015), ImageNet-64 (Deng et al., 2009) and LSUN-Bedroom (Yu et al., 2015). Our searched solver schedules outperform the SOTA diffusion sampler (Lu et al., 2022b; Zhao et al., 2023) by a large margin with very few NFE, e.g., 6.86 v.s. 23.44 on CIFAR-10 with only 5 NFE. We further apply $S^3$ to Stable-Diffusion (Rombach et al., 2022) models, achieve 2× (from 10 NFE to 5 NFE) acceleration without sacrificing the performance on text-to-image generation task on MS-COCO 256×256 dataset (Lin et al., 2014). Based on the experimental results, we summarize some knowledge to guide future schedule design.

## 2 RELATED WORK

### 2.1 DIFFUSION PROBABILISTIC MODEL

Diffusion Probabilistic Models (DPMs) (Sohl-Dickstein et al., 2015; Ho et al., 2020; Song et al., 2020b) are used to construct the probability distribution $q(x_0)$ of a $D$-dimension random variable $x_0 \in \mathbb{R}^D$. DPMs define a forward diffusion process to add noise to the random variable $x_0$ (Song et al., 2020b) progressively:

$$\mathrm{d}x_t = f(x_t, t)\mathrm{d}t + g(t)\mathrm{d}w_t, \tag{1}$$

where $x_t$ stands for the marginal distribution at time $t$ for $t \in [0, T]$, and $x_0$ obeys the target distribution $q(x_0)$. $w_t$ is the standard Wiener process. When $f(x_t, t)$ is a affine form of $x_t$, the marginal

distribution of $x_t$ given $x_0$ can be written as (Song et al., 2020b):

$$q(x_t|x_0) = \mathcal{N}(x_t|\alpha_t x_0, \sigma_t^2 \boldsymbol{I}), \tag{2}$$

where $\alpha_t^2/\sigma_t^2$ is called *signal-to-noise ratio* (SNR), which is a strictly decreasing function of $t$ and approximately equals 0 when $t = T$. The forward diffusion process Eq. (1) has a reverse probability flow ODE (Song et al., 2020b;a):

$$\mathrm{d}x_t = [f(t)x_t - \frac{1}{2}g^2(t)\nabla_x \mathrm{log}q(x_t)]\mathrm{d}t, \tag{3}$$

where $x_T \sim \mathcal{N}(x_t|0, \boldsymbol{I})$. The ODE has two equivalent forms w.r.t. noise prediction network $\epsilon_\theta$ and data prediction network $x_\theta$, which are more commonly used for fast sampling:

$$\mathrm{d}x_t = (f(t)x_t + \frac{g^2(t)}{2\sigma_t}\epsilon_\theta(x_t, t))\mathrm{d}t, \tag{4}$$

$$\mathrm{d}x_t = ((f(t) + \frac{g^2(t)}{2\sigma_t^2})x_t - \frac{\alpha_t g^2(t)}{2\sigma_t^2}x_\theta(x_t, t))\mathrm{d}t. \tag{5}$$

It can be proved that the marginal distribution of $x_t$ in Eq. (1) equals the distribution of $x_t$ in Eq. (3) (Anderson, 1982). Thus, sampling from DPMs can be viewed as solving the above ODEs.

## 2.2 Training-Free Samplers

One major drawback of DPMs is the large number of discretized timesteps needed for numerical solvers to ensure the sample quality. To handle such an issue, training-free samplers for the ODE Eq. (3) are proposed to achieve quicker convergence (Song et al., 2020b;a; Liu et al., 2022; Zhang & Chen, 2022; Lu et al., 2022a;b; Zhao et al., 2023).

### 2.2.1 Low Order Samplers

The early samplers can be viewed as low-order solvers of the diffusion ODE (Song et al., 2020a; Liu et al., 2022). **DDIM** (Song et al., 2020a) proposes the following formula for the update of one step: $x_t = \sqrt{\alpha_t}\frac{x_s - \sqrt{1-\alpha_s}\epsilon_\theta(x_s, s)}{\sqrt{\alpha_s}} + \sqrt{1-\alpha_t}\epsilon_\theta(x_s, s)$. **PNDM** (Liu et al., 2022) proposes to replace the neural term $\epsilon_\theta(x_s, s)$ in the DDIM formula with a new term $\frac{1}{24}(55\epsilon_s - 59\epsilon_{s-\delta} + 37\epsilon_{s-2\delta} - 9\epsilon_{s-3\delta})$ inspired by Adams method. The DDIM formula is a 1-st order discretization to the diffusion ODE Eq. (3) and thus is not efficient enough due to the inadequate utilization of high-order information.

### 2.2.2 Exponential-Integral-Based Samplers

The solution $x_t$ of the diffusion ODE Eq. (3) at timestep $t$ from timestep $s$ can be analytically computed using the following exponentially weighted integral (Lu et al., 2022a)

$$x_t = \frac{\alpha_t}{\alpha_s}x_s - \alpha_t \int_{\lambda_s}^{\lambda_t} e^{-\lambda}\epsilon_\theta(x_\lambda, \lambda)\mathrm{d}\lambda. \tag{6}$$

The linear term $\frac{\alpha_t}{\alpha_s}x_s$ can be accurately computed and the high order numerical methods can be applied to the integral term $\alpha_t \int_{\lambda_s}^{\lambda_t} e^{-\lambda}\epsilon_\theta(x_\lambda, \lambda)\mathrm{d}\lambda$, improving the sample speed dramatically than low-order formulas. The following methods apply different strategies to estimate the integral term.

**DEIS** (Zhang & Chen, 2022) uses Lagrange outer interpolation to estimate the integrand. Then, the exponential integral can be computed approximately since the integral of a polynomial can be easily computed. Additionally, DEIS computes this integral in $t$ domain rather than $\lambda$.

**DPM-Solver** (Lu et al., 2022a) utilizes the Taylor expansion of the exponential integral as follows to estimate this term.

$$x_t = \frac{\alpha_t}{\alpha_s}x_s - \alpha_t \sum_{n=0}^{k-1}\epsilon_\theta^{(n)}(x_{\lambda_s}, \lambda_s)\int_{\lambda_s}^{\lambda_t} e^{-\lambda}\frac{(\lambda-\lambda_s)^n}{n!}\mathrm{d}\lambda + \mathcal{O}((\lambda_t - \lambda_s)^{k+1}). \tag{7}$$

Additionally, Lu et al. (2022a) proposes to discretize the timesteps with uniform logSNR, which has empirically better performance. **DPM-Solver++** (Lu et al., 2022b) introduces different strategies under the same Taylor expansion framework, e.g., multi-step solver and data prediction neural term.

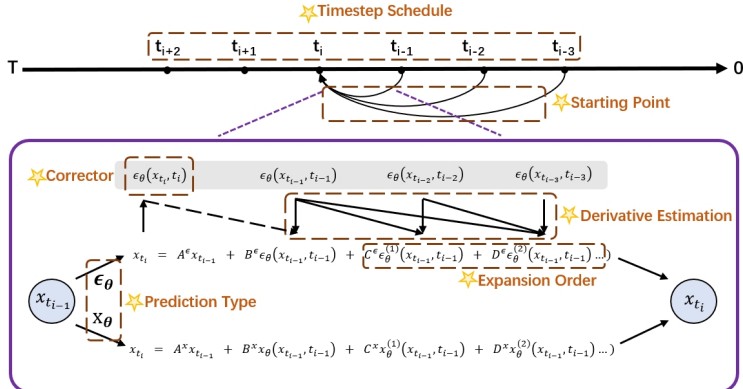

Figure 1: The sampling process of USF and all searchable strategies.

**UniPC** (Zhao et al., 2023) introduces correcting strategy to the Taylor expansion framework. After computing $x_t$ through Eq. (7), UniPC uses the newly computed function evaluation $\epsilon_\theta(x_t, \lambda_t)$ to correct the value of $x_t$, which does not consume extra NFE as $\epsilon_\theta(x_t, \lambda_t)$ is calculated only once.

## 2.3 AutoML

AutoML methods aim to automatically decide the optimal configurations for machine learning systems. The decidable elements in AutoML works include training hyperparameters (Jaderberg et al., 2017), model selection (Yang et al., 2019; Liu et al., 2023), neural architecture (Zoph & Le, 2016), etc. Predictor-based search methods (Snoek et al., 2012) are widely used in the AutoML field to accelerate the optimization. In predictor-based search, a predictor is trained with evaluated configuration-objective pairs, and then used to predict the objectives for new configurations.

## 3 A Unified Sampling Framework

In this section, we propose a unified sampling framework called USF based on exponential integral in the $\lambda$ domain. We introduce our framework by describing the steps of solving the diffusion ODE Eq. (3) and listing all optional strategies and tunable hyperparameters of the sampling process.

### 3.1 Solving Strategy

To solve the diffusion ODE Eq. (3) without closed-form solutions, one should discretize the continuous time into timesteps and calculate the trajectory values at each timestep. Therefore, the first step of solving the diffusion ODE is to determine a **discretization scheme** $[t_0, t_1, \cdots, t_N]$, where $\varepsilon = t_0 < t_1 < \cdots < t_N = T$. Then consider a timestep $t \in [t_0, \cdots, t_{N-1}]$, exponential integral has been validated to be an effective method of computing $x_t$ due to the semi-linear property (Zhang & Chen, 2022; Lu et al., 2022a;b; Zhao et al., 2023), especially in the $\lambda$ domain. We follow the Taylor expansion of Eq. (6) in $\lambda$ domain (Lu et al., 2022a) to estimate the integral term, and summarize the strategies that can be applied to numerically compute the expansion as follows.

**Prediction type of neural network**. Typically, the neural network is trained with denoising objective (Ho et al., 2020) and the corresponding diffusion ODE writes as Eq. (4). It can be proved that Eq. (6) is the accurate solution of Eq. (4) (see App. H). Denote that $s$ is another timestep before $t$ and $h = \lambda_t - \lambda_s$, the Taylor expansion of Eq. (6) can be written as (Lu et al., 2022a):

$$x_t = \frac{\alpha_t}{\alpha_s} x_s - \sigma_t \sum_{k=0}^{n} h^{k+1} \varphi^\epsilon_{k+1}(h) \epsilon_\theta^{(k)}(x_s, s) + \mathcal{O}(h^{n+2}), \tag{8}$$

where $\varphi^\epsilon_k$ satisfies $\varphi^\epsilon_{k+1}(h) = \frac{\varphi^\epsilon_k(h) - 1/k!}{h}$, and $\varphi^\epsilon_0(h) = e^h$. Alternatively, the neural network can be a *data prediction model* $x_\theta(x_t, t)$, whose relationship with the noise prediction model $\epsilon_\theta(x_t, t)$ is given by $x_\theta(x_t, t) := \frac{x_t - \sigma_t \epsilon_\theta(x_t, t)}{\alpha_t}$ (Kingma et al., 2021; Lu et al., 2022b). The corresponding diffusion ODE w.r.t. data prediction model writes as Eq. (5), whose Taylor expansion is given below:

$$x_t = \frac{\sigma_t}{\sigma_s} x_s + \alpha_t \sum_{k=0}^{n} h^{k+1} \varphi^x_{k+1}(h) x_\theta^{(k)}(x_s, s) + \mathcal{O}(h^{n+2}), \tag{9}$$

where $\varphi_k^x$ satisfies $\varphi_{k+1}^x(h) = \frac{1/k! - \varphi_k^x(h)}{h}$, and $\varphi_0^x(h) = e^{-h}$. See App. H for detailed derivations. Since the linear term and the integral to estimate are different between the two prediction types, the numerical solutions of Eq. (8) and Eq. (9) are essentially different except for $n = 1$.

**Starting point** $s$. $s$ can be any timestep larger than $t$. Multistep solvers (Lu et al., 2022b; Zhao et al., 2023) always choose the previous timestep of $t$ while single-step solvers (Lu et al., 2022a) choose more distant timesteps.

**Order of the Taylor expansion**. To compute the Taylor expansion Eq. (8) or Eq. (9), the number of order retained should be decided. Low retained order may result in higher truncation error. However, the estimation of high-order derivatives need extra information from other timesteps other than the starting point $s$, which may not be accurate enough for calculation. Therefore, the choice of Taylor expansion order may have a strong influence on the sample quality.

**Derivative estimation method**. After determining the prediction type and the retained order $n$, the last thing is to numerically calculate all unknown derivative terms $\epsilon_\theta^{(k)}(x_s, s)$ (take noise prediction model as an example), where $k \leq n$, to obtain the final solution $x_t$. The strategies of this component are very flexible since many off-the-shelf numerical differential methods can be used. To simplify the workflow, we utilize the system of linear equations consisting of $m \geq k$ Taylor expansions from the starting point $s$ to other timesteps, the $k$-order derivative can be approximately calculated by eliminating derivatives of other orders (Zhao et al., 2023), which we call *m-th Taylor-difference method* (see Def. B.1 for its formal definition). Inspired by the 2-nd solver of Lu et al. (2022a;b), we note that for a derivative estimation result $\widetilde{\epsilon}_\theta^{(k)}(x_s, s)$, rescaling the estimated value by multiplying a coefficient $1 + R(h)$ has the potential to further correct this numerical estimation. To sum up, we formulate the estimation of $k$-th order derivative as the following two steps: **(1)**. choose at least $k$ other timesteps $t_i \neq s, i \in [1, \cdots, k]$ with pre-computed function evaluation $\epsilon_\theta(x_{t_i}, t_i)$ to calculate the estimation value through *Taylor-difference method*, and **(2)**. rescale the estimation value with a coefficient $1 + R(h)$. In the search space used in our experiments, we allow low-order Taylor-difference estimation to be taken for the 1-st derivative in high-order Taylor expansions (i.e., $m$ can be smaller than $n$) and design five types of scale coefficients $R_i(h), i = 0, \cdots, 4$ for the 1-st derivative. See App. B for details.

After calculating $x_t$, **correctors** can be used to improve the accuracy of $x_t$ through involving the newly computed $\epsilon_\theta(x_t, t)$ in the re-computation of $x_t$ (Griffiths & Higham, 2010). Specifically, the Taylor expansion Eq. (8) or Eq. (9) is estimated again with $t$ as one other timestep for derivative estimation. Typically, $\epsilon_\theta(x_t, t)$ also have to be evaluated again after recomputing $x_t$ and the correcting process can be iteratively conducted. However, in practice, re-evaluations and iterative correctors are not used to save NFE budgets. Since our goal is to improve the sample quality with very few NFE, we follow Zhao et al. (2023) to use no additional neural inference for correctors.

In summary, all decidable components in our framework are: **(1)**. timestep discretization scheme, **(2)**. prediction type of the neural term, **(3)**. starting point of the current step, **(4)**. retained order of Taylor expansion, **(5)**. derivative estimation methods, and **(6)**. whether to use corrector. Our sampling framework is summarized in Alg. 1 and demonstrated in Fig. 1.

## 3.2 SOLVER SCHEDULE: PER-STEP SOLVING STRATEGY SCHEME

In this section, we demonstrate the phenomenon that the proper solving strategies appropriate for different timesteps are different. We construct a 'ground-truth' trajectory $c = [x_{t_0}, x_{t_1}, \cdots, x_{t_N}]$ using 1000-step DDIM (Song et al., 2020a) sampler (i.e., $N = 1000$). We choose a serials of target timesteps $[t_{tar_0}, \cdots, t_{tar_S}]$ where $t_{tar_i} \in [t_0, \cdots, t_N]$, and estimate the value $\tilde{x}_{t_{tar_i}}$ with $k$ previous timesteps $[t_{s_1}, \cdots, t_{s_j}, \cdots, t_{s_k}]$, where $s_j \in [1, N]$ and $t_{s_j} > t_{tar_i}$. Then we calculate the distance between the ground-truth trajectory value and the estimated value, given by $L(t_{tar_i}) = |\tilde{x}_{t_{tar_i}} - x_{t_{tar_i}}|$. We compare different solving strategies by showing their $L(t_{tar_i})$-$t_{tar_i}$ curve in Fig. 2. The key observation is that **the suitable strategies for most components varies among timesteps**. For example, the loss of the 3-rd solver with noise prediction network is smaller than the 2-nd solver at 400-700 timesteps but larger at the rest timesteps (see the green and red curves of 'Orders and Prediction Types').

This phenomenon reveals the potential of using different solving strategies in different timesteps. We call the decision sequence of solving strategies for every timestep *solver schedule*.

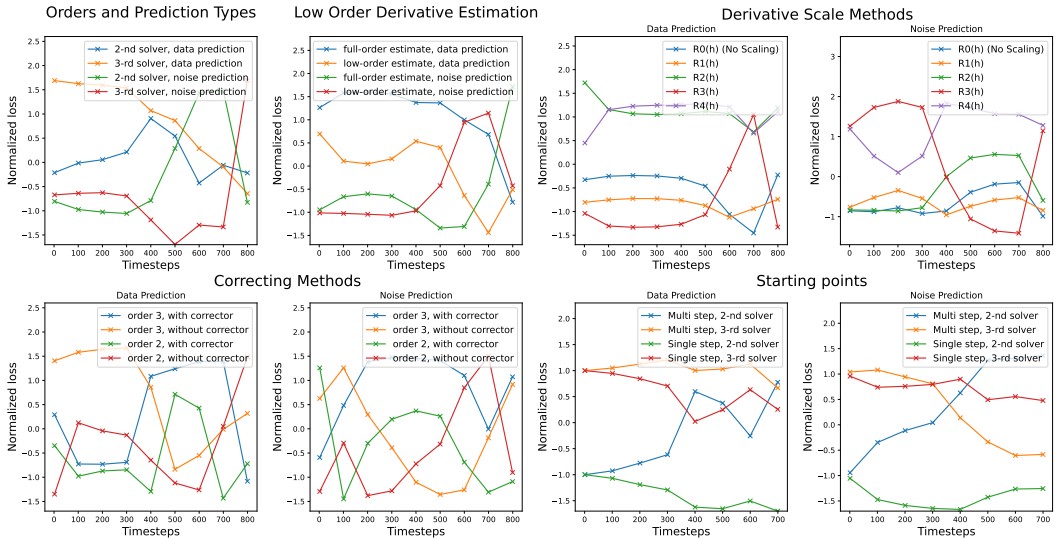

Figure 2: Losses with ground-truth trajectory using different strategies. **(i)**. For the losses in 'Orders and Prediction Types', we choose different expansion orders and prediction types for every update. **(ii)**. For the losses in 'Low Order Derivative Estimation', we use the 3-rd Taylor expansion with both prediction types and choose whether to apply the 1-st Taylor-difference method to estimate the 1-st order derivative rather than the 2-nd Taylor-difference method. **(iii)**. For the losses in 'Correcting Methods', we choose whether to use correctors after the update under different expansion orders and prediction types. **(iv)**. For the losses in 'Starting points', we choose different indices of the starting points, including 'Single step' (all additional timesteps are between the starting point and the target point) and 'Multi step' (all additional timesteps are larger than the starting point). **The ranks of all strategies for all components are not constant at different timesteps.**

### 3.3 ADVANTAGES OF THE FRAMEWORK

The contribution of USF lies in two ways. First, USF gives a unified perspective for all sampling methods based on exponential integral. Second, USF allows different solving strategies in each timestep, making the sampler more flexible. To demonstrate the universality and reliability of USF, we show in Tab. 1 that almost all existing SOTA samplers based on exponential integral can be incorporated by it through assigning corresponding strategies (more derivations can be found in App. E). Note that since DEIS (Zhang & Chen, 2022) computes the integral in the $t$ domain, it is essentially different from other methods. However, when applying the core numerical method in DEIS to the $\lambda$ domain, we show that it can still be incorporated in USF (the last row in Tab. 1).

## 4 SEARCH FOR SOLVER SCHEDULE THROUGH $S^3$

### 4.1 PROBLEM DEFINITION

Denote the $s_i$ as the numerical solver to compute $x_{t_i}$, we formulate the problem of deciding the optimal solver schedule as below:

$$\underset{\substack{M \le L, \\ s_1, s_2, \cdots, s_M, \\ t_1, t_2, \cdots, t_M}}{\arg\min} \quad \mathbb{I}_{\mathcal{F}} * \mathcal{F}([(s_1, t_1), (s_2, t_2), \cdots, (s_M, t_M)]), \text{s.t.} M \le C,$$

where $C$ is the given NFE budget, $[(s_1, t_1), (s_2, t_2), \cdots, (s_M, t_M)]$ is the solver schedule s, and $\mathcal{F}$ is an arbitrary evaluation metric of diffusion models, and $\mathbb{I}_{\mathcal{F}}$ is an indicator coefficient which equals to 1 when smaller $\mathcal{F}$ indicates better performance and equals to $-1$ otherwise.

As discussed in Sec. 3.2, the best solving strategies for different timesteps are different, which leads to a search space that grows exponentially with the number of steps. Additionally, the decision of one component may be coupled with the decision of other components. For example, as demonstrated in Fig. 2, low-order estimation of derivatives almost consistently outperforms full-order estimation method with *data prediction* network (see the orange and blue curves of 'Low Order Derivative

Table 1: Relationship between USF and existing SOTA methods. The value of 'starting point' represents the index difference between the target point and the starting point. All methods use Taylor-difference to estimate derivatives or all orders. The formulas in 'Scale' are the coefficients for scaling the 1-st derivative. $h'$ equals $h$ if prediction type is noise and equals $-h$ if it is data.

| Method | Prediction | Taylor order | Starting point | Scale | Corrector |
|---|---|---|---|---|---|
| DPM-Solver-2S | Noise | 1,2 | -1,-2 | $\frac{\frac{h}{2}*(e^h-1)}{e^h-h-1}$ | None |
| DPM-Solver-3S | Noise | 1,2,2 | -1,-2,-3 | None | None |
| DPM-Solver++(2M) | Data | 2 | -1 | $\frac{\frac{h}{2}*(1+e^{-h})}{e^{-h}+h-1}$ | None |
| UniPC-2-$B_1(h)$ | Noise/Data | 2 | -1 | $\frac{\frac{h'^2}{2}}{e^{h'}-h'-1}$ | UniC-2 |
| UniPC-2-$B_2(h)$ | Noise/Data | 2 | -1 | $\frac{\frac{h'}{2}(e^{h'}-1)}{e^{h'}-h'-1}$ | UniC-2 |
| UniPC-$p$ $(p>2)$ | Noise/Data | $p$ | -1 | None | UniC-$p$ |
| DEIS-$p$ (in the $\lambda$ domain) | Noise | $p$ | -1 | None | None |

Estimation') while worse than it with *noise prediction* network at 500-700 timesteps (see the red and green curves). Therefore, well-performing solver schedules are hard to design manually.

Inspired by AutoML works (Real et al., 2019; Ning et al., 2020; Liu et al., 2023), we attempt to automatically search for solver schedules with good performance under certain NFE budgets. However, the search space is extremely large, and evaluating a solver schedule is also time-consuming since the standard calculation processes of most metrics need to generate a large number of images (Heusel et al., 2017). Thus, random search with a regular metric evaluation method is inefficient in this case.

### 4.2 PREDICTOR-BASED MULTI-STAGE SEARCH FOR SOLVER SCHEDULE

To surmount the aforementioned challenges and accelerate the search process, we propose $S^3$, a predictor-based multi-stage evolutionary method. Specifically, we train a light predictor to predict the performance of a solver schedule and use it to guide an evolutionary search process to sample solver schedules. We can evaluate new solver schedules in the evolutionary search with negligible cost with the help of the predictor. making the search process much more efficient.

But training the predictor still needs a bunch of evaluated schedules, which is time consuming. To decrease this part of cost, we propose the multi-stage workflow $S^3$. In the early stage of the evolutionary search, only the sub-space adjacent to the current population can be explored. Therefore, the predictor is not required to generalize to the whole search space and can be trained with a small amount of data. As the population expands, generalization ability to new sub-spaces is needed, and thus the predictor should be trained with more sampled schedules. Moreover, the training data of the predictor should also be selected carefully and efficiently.

Based on the above intuition, we propose the multi-stage workflow. Our workflow is summarized in Alg. 2 and demonstrated in App. D.2, which contains three steps in every loop. **(1)**. A set of solver schedules are sampled from the search space through evolutionary search. The performances needed to guide the evolutionary search are calculated by the predictor except for the first loop in which all performances are truly evaluated. **(2)**. The true metrics of all sampled solver schedules are calculated. **(3)**. All evaluated schedule-performance data pairs are used to update the weights of the predictor. Compared to the single-stage method, which evaluates the true performance of all schedules in one go and trains a predictor to search in the whole space, $S^3$ accelerates the exploration of schedules which have to be truly evaluated.

## 5 EXPERIMENTS

We choose several SOTA diffusion samplers, including DPM-Solver (Lu et al., 2022a), DPM-Solver++ (Lu et al., 2022b), and UniPC (Zhao et al., 2023) as baseline methods. We use FID↓ (Heusel et al., 2017) as the evaluation metric. Our method is validated on CIFAR-10, CelebA, ImageNet-64, ImageNet-128 (guided), ImageNet-256 (guided), and LSUN-bedroom datasets and outperforms baseline methods by a large margin on all of them. We further apply $S^3$ to text-image generation task with Stable Diffusion pre-trained models (Rombach et al., 2022) on MS-COCO2014 validation set. The models we use for all experiments are listed in App. F.1.

Table 2: FIDs of the searched solver schedules on unconditional and label-guided generation tasks.

| Dataset | Method | NFE | | | | | | |
|---------|--------|-----|-----|-----|-----|-----|-----|-----|
| | | 4 | 5 | 6 | 7 | 8 | 9 | 10 |
| **CIFAR-10** | Baseline-W(S) | 255.21 | 288.12 | 32.15 | 14.79 | 22.99 | 6.41 | 5.97 |
| | Baseline-W(M) | 61.13 | 33.85 | 20.84 | 13.89 | 10.34 | 7.98 | 6.76 |
| | Baseline-B | 57.52 | 23.44 | 10.33 | 6.47 | 5.16 | 4.30 | 3.90 |
| | Ours | **11.50** | **6.86** | **5.18** | **3.81** | **3.41** | **3.02** | **2.69** |
| **CelebA** | Baseline-W(S) | 321.39 | 330.10 | 52.04 | 17.28 | 16.99 | 10.39 | 6.91 |
| | Baseline-W(M) | 31.27 | 20.37 | 14.18 | 11.16 | 9.28 | 8.00 | 7.11 |
| | Baseline-B | 26.32 | 8.38 | 6.72 | 6.72 | 5.17 | 4.21 | 4.02 |
| | Ours | **12.31** | **5.17** | **3.65** | **3.80** | **3.62** | **3.16** | **2.73** |
| **ImageNet-64** | Baseline-W(S) | 364.60 | 366.66 | 72.47 | 47.84 | 54.21 | 28.22 | 27.99 |
| | Baseline-W(M) | 93.98 | 69.08 | 50.35 | 40.99 | 34.80 | 30.56 | 27.96 |
| | Baseline-B | 76.69 | 61.73 | 42.81 | 31.76 | 26.99 | 23.89 | 24.23 |
| | Ours | **33.84** | **24.95** | **22.31** | **19.55** | **19.19** | **19.09** | **16.68** |
| **LSUN-Bedroom** | Baseline-W(M) | 44.29 | 24.33 | 15.96 | 12.41 | 10.87 | 9.99 | 8.89 |
| | Baseline-B | 22.02 | 17.99 | 12.43 | 10.79 | 9.92 | 9.11 | 8.52 |
| | Ours | **16.45** | **12.98** | **8.97** | **6.90** | **5.55** | **3.86** | **3.76** |
| **ImageNet-128** | Baseline-W(M) | 32.08 | 15.39 | 10.08 | 8.37 | 7.50 | 7.06 | 6.80 |
| | Baseline-B | 25.77 | 13.16 | 8.89 | 7.13 | 6.28 | 6.06 | 6.03 |
| | Ours | **18.61** | **8.93** | **6.68** | **5.71** | **5.28** | **4.81** | **4.69** |
| **ImageNet-256** | Baseline-W(M) | 80.46 | 54.00 | 38.67 | 29.35 | 22.06 | 16.74 | 13.66 |
| | Baseline-B | 51.09 | 27.71 | 17.62 | 13.19 | 10.91 | 9.85 | 9.31 |
| | Ours | **33.84** | **19.06** | **13.00** | **10.31** | **9.72** | **9.06** | **9.06** |

We evaluate 9 common settings in total of these SOTA methods. We list the *best* (Baseline-B in all tables) and the *worst* (Baseline-W in all tables) results of all baseline methods we evaluate under the same NFE budget. For the worst baseline, we distinguish singlestep methods (S) and multistep methods (M) since the former is usually significantly worse with meager budgets. Detailed settings and full results can be found in App. F.3 and App. G.1, correspondingly.

## 5.1 MAIN RESULTS

We list the results on unconditional datasets and with classifier-guidance in Tab. 2 and the results with Stable Diffusion models in Tab. 3. The key takeaways are: (1) $S^3$ **can achieve much higher sample quality than baselines, especially with very few steps.** Our searched schedules outperform all baselines across all budgets and datasets by a large margin. On text-to-image generation with Stable Diffusion models, $S^3$ achieves comparable FIDs with only 5 NFE compared to baselines with 10 NFE, bringing a $2\times$ acceleration ratio. Remarkably, our method achieves a significant boost under very tight budgets (like 11.50 v.s. 57.52 on CIFAR-10, 33.84 v.s. 76.69 on ImageNet-64), making it feasible to sample with very low NFEs. (2) **Choosing proper settings for existing methods is important.** We can see that in many cases, the performances of the best baseline and the worst baseline are significantly different from each other, even both of which are under the SOTA frameworks. Additionally, the best baselines on different datasets or even under different NFE budgets are not consistent. It indicates the necessity of a careful selection of solver settings and the significance of our USF since it is proposed to present all adjustable settings systematically.

## 5.2 ABLATION STUDY: REDUCE SEARCH OVERHEAD

The main overhead of our method comes from evaluating the true performance of solver schedules since the predictor is much lighter than diffusion U-Nets and has negligible overheads. The time consumed is proportional to the number of generated images used for metric calculation. Fewer evaluation images lead to higher variance and lower generalization ability to other texts and initial noises, thus may cause worse performance. In this section, we ablate this factor to give a relationship

Table 3: FID results on MS-COCO 256×256. Ours-500/Ours-250 stand for using 500/250 images to calculate FID when searching.

| Method | NFE | | | | | | |
|---|---|---|---|---|---|---|---|
| | 4 | 5 | 6 | 7 | 8 | 9 | 10 |
| Baseline-W(S) | 161.03 | 156.72 | 106.15 | 75.28 | 58.54 | 39.26 | 29.54 |
| Baseline-W(M) | 30.77 | 22.71 | 19.66 | 18.45 | 18.00 | 17.65 | 17.54 |
| Baseline-B | 24.95 | 20.59 | 18.80 | 17.83 | 17.54 | 17.42 | 17.22 |
| Ours | **22.76** | **16.84** | **15.76** | 14.77 | **14.23** | **13.99** | **14.01** |
| Ours-500 | 24.47 | 17.72 | 15.71 | **14.60** | 14.47 | 14.15 | 14.27 |
| Ours-250 | 23.84 | 18.27 | 17.29 | 14.90 | 15.50 | 14.12 | 14.31 |

between overhead and performance. We choose the text-to-image task because the text condition is far more flexible than label condition or no condition and thus more difficult to generalize from a small number of samples to other samples. We validate $S^3$ with 250 and 500 generated images rather than the default setting 1000, and our results are shown in the last two rows in Tab. 3. The performance under almost all NFE budgets becomes worse as the number of generated samples decreases. Moreover, some FIDs under higher budgets are even higher compared to lower budgets due to the high variance caused by lower sample numbers. Remarkably, the search results can still outperform baseline methods, indicating the potential of $S^3$.

## 5.3 ANALYSIS AND INSIGHTS

We give some empirical insights in this section based on the pattern of our searched schedules. Our general observation is that most components of searched schedules show different patterns on different datasets. We analyze the patterns of different components as follows.

**Timesteps.** For low resolution datasets, more time points should be placed when $t$ is small. Unlike the default schedule "logSNR uniform", we suggest putting more points at $0.2 < t < 0.5$ rather than putting too much at $t < 0.05$. For high resolution datasets, we recommend a slightly smaller step size at $0.35 < t < 0.75$ on top of the uniform timestep schedule.

**Prediction Types.** We find that data prediction can outperform noise prediction by a large margin in the following cases: 1. the NFE budget is very low (i.e., 4 or 5), 2. the Taylor order of predictor or corrector is high and 3. on low resolution datasets. But for the rest cases, their performance has no significant difference. Noise prediction can even outperform data prediction sometimes.

**Derivative Estimation Methods.** We find that low order derivative estimation is preferred for high Taylor order expansion when step size is very high. Oppositely, full-order estimation is more likely to outperform low-order estimation when step size is not so large. For derivative scaling functions, we find they are often used in all searched schedules. But the pattern varies among datasets.

**Correctors.** Though previous work validates the effectiveness of using correctors Zhao et al. (2023), we find that not all timesteps are suitable for correctors. For large guidance scale sampling, corrector is not usually used. But in general, corrector is more likely to bring positive impact on the performance.

## 6 CONCLUSION AND FUTURE WORK

In this paper, we propose a new sampling framework, USF, for systematical studying of solving strategies of DPMs. We further reveal that suitable strategies at different timesteps are different, and thus propose to search solver schedules with the proposed framework in the predictor-based multi-stage manner. Experiments show that our proposed method can boost the sample quality under a very tight budget by a large margin, making it feasible to generate samples with very few NFE.

Although we propose a general framework for sampling based on exponential integral, we prune the search space empirically to avoid excessive search overheads. Exploring the strategies not used in this work could be a valuable direction for future research. Besides, our method has additional overhead related to the evaluation speed and the number of sampled schedules. Therefore, fast evaluation methods and efficient search methods are worth studying.

ACKONWLEDGEMENT

This work was supported by National Natural Science Foundation of China (No. 62325405, 62104128, U19B2019, U21B2031, 61832007, 62204164), Tsinghua EE Xilinx AI Research Fund, and Beijing National Research Center for Information Science and Technology (BNRist). We thank Cheng Lu and Prof. Jianfei Chen for valuable suggestions and discussion, Yizhen Liao for providing some experimental results and reviewing the revision in the rebuttal phase. We thank for all the support from Infinigence-AI.

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

## A  DEFINITIONS AND ASSUMPTIONS

In this section we list some necessary definitions and assumptions for the discussion follows. We first define the convergence order of an arbitrary estimation $\tilde{x}$.

**Definition A.1.** Suppose $h$ is a given value, $\tilde{x}(h)$ is an estimation of the ground truth value $x$. If $\exists c$, such that $\forall h$, $|\tilde{x}(h) - x| \leq ch^p$, then we call $\tilde{x}(h)$ is $p$-order converge to $h$.

The defined $p$-order convergence can also be given as $\tilde{x}(h) = x + \mathcal{O}(h^p)$.

Then we list several regularity assumptions for the neural network $\epsilon_\theta$, $x_\theta$, and the hyperparameters of DPMs.

**Assumption A.1.** $\epsilon_\theta(x, t)$ and $x_\theta(x, t)$ are Lipschitz w.r.t. $x$ with a Lipschitz constant $L$

**Assumption A.2.** The $i$-th derivatives $\frac{d^i \epsilon_\theta(x_t, \lambda_t)}{d\lambda^i}$ and $\frac{d^i x_\theta(x_t, \lambda_t)}{d\lambda^i}$ exist and are continuous for all $1 \leq i \leq M$, where $M$ is an large enough value.

**Assumption A.3.** $\exists B$ such that $\forall t, s \in (t_\varepsilon, T]$ and $t < s$, $\frac{\sigma_t}{\sigma_s} < \frac{\alpha_t}{\alpha_s} \leq B$.

## B  SEARCH SPACE DETAILS

In this section, we detail the searchable strategies for all components of the search space we use for experiments in Sec. 5. We also try several different designs and list the results in App. G.4.

**Timestep Schedule** For all experiments, we set the start timestep $t_N = T$ following the conclusion in Lin et al. (2023) that setting $t_N \neq T$ can be harmful to the sample quality. For the last timestep $t_0$, previous works usually set it to 0.001 or 0.0001 empirically (Lu et al., 2022a;b). We find that the value of $t_0$ has a large impact on the sample quality since low $t_0$ can cause numerical instability and high $t_0$ contains more noise. Therefore, we set $t_0$ to be searchable in the interval [0.0001, 0.0015]. For other $t_i$ where $i \in [1, N-1]$, all values between $t_0$ and $t_N$ are optional.

**Prediction Type.** For the experiments in Sec. 5, we simply set data prediction and noise prediction as the two only strategies that can be applied for one-step update. We further try interpolation between the two forms: given $x_t^\epsilon$ calculated by Eq. (8) and $x_t^x$ calculated by Eq. (9), we set the final solution $x_t$ to be $ax_t^\epsilon + (1-a)x_t^x$. The results of this adjusted search space are shown in Tab. 19.

**Starting Point.** We observe that in most cases, singlestep solvers perform worse than multistep solvers. Therefore, we simply set the starting point to be the previous timestep $x_{t_{i+1}}$ of the target point $x_{t_i}$ for all experiments in Sec. 5. To have a better understanding of the singlestep approach, we further set the starting point searchable with two candidates $x_{t_{i+1}}$ and $x_{t_{i+2}}$ for every $x_{t_i}$, and list the search results with this new search space in Tab. 18.

**Taylor expansion order.** As discussed in the previous work Zhao et al. (2023), too high solver order is detrimental to the solution, possibly because of the numerical instability of high-order derivatives. So, we set the candidates of Taylor expansion order as 1, 2, 3, and 4.

**Derivative Estimation Method.** As discussed in Sec. 3.1, we divide the estimation of derivatives into two steps: Taylor-difference and scaling operation. For the Taylor-difference method, the main idea is to utilize Taylor expansions from other points to the target points and eliminate all derivatives of other orders to preserve the derivative needed to be computed. We give the formal definition here.

**Definition B.1.** Given the function value of the target point $f(x_t, t)$, and $m$ additional points $f(x_{t_1}, t_1), f(x_{t_2}, t_2), \cdots, f(x_{t_m}, t_m)$ where $t_i = t + r_i h$, then $f^{(k)}(x_t, t)$ can be estimated by $\boldsymbol{D}^T \boldsymbol{a}/h^k$, where $\boldsymbol{D}$ is a $m$-dimension vector given by $\boldsymbol{D} = [f(x_t, t) - f(x_{t_1}, t_1), f(x_t, t) -$

$f(x_{t_2}, t_2), \cdots, f(x_t, t) - f(x_{t_m}, t_m)]$. And $\boldsymbol{a} = [a_1, a_2, \cdots, a_m]$ is the solution vector of the linear equation system $\boldsymbol{Ca} = \boldsymbol{b}$, where $\boldsymbol{C}$ is a square matrix with $m$ rows and columns satisfying $\boldsymbol{C}_{ij} = \frac{r_j^i}{i!}$ for all integers $i, j \in [1, m]$, and $\boldsymbol{b}$ is a $m$-dimension vector satisfying $\boldsymbol{b}_i = \mathbb{I}_{i==k}$ for all integers $i \in [1, m]$.

The convergence order $p$ of Taylor-difference estimation is given by $p = m - k + 1$. The proof is simple by substituting the Taylor expansion of all $f(x_{t_i}, t_i)$ to $f(x_t, t)$ into the estimation formula.

*Proof.* Write the estimated value $\tilde{f}^{(k)}(x_t, t)$ through Def. B.1:

$$
\begin{aligned}
\tilde{f}^{(k)}(x_{t_0}, t_0) =& \frac{1}{h^k} \sum_{i=1}^{m} a_i(f(x_t, t) - f(x_{t_i}, t_i)) \\
=& \frac{1}{h^k} \sum_{i=1}^{m} a_i(r_i h f^{(1)}(x_{t_0}, t_0) + \frac{(r_i h)^2 f^{(2)}(x_{t_0}, t_0)}{2} + \\
& \cdots + \frac{(r_i h)^m f^{(m)}(x_{t_0}, t_0)}{n!} + \mathcal{O}(h^{m+1})) \\
=& f^{(k)}(x_{t_0}, t_0) + \mathcal{O}(h^{m-k+1})
\end{aligned}
\tag{10}
$$

$\square$

The last equation holds because $a_i$ is a constant and independent of $h$. If we consider the error of all involved points $\tilde{x}_{t_i}$ and $f$ satisfy the regularity assumptions in App. A, then the convergence order should be $p = min(m, l_i) - k + 1$, where $l_i$ is the convergence order of $\tilde{x}_{t_i}$. According to Asm. A.1, the convergence order of $f(\tilde{x}_{t_i}, t_i)$ equals the convergence order of $\tilde{x}_{t_i}$. By substituting the convergence order of $f$ into Eq. (10), we can get a similar proof.

The derivation of the last equation utilizes $\boldsymbol{Ca} = \boldsymbol{b}$ that $\sum_{i=1}^{m} a_i \frac{r_i^k}{k!} = \mathbb{I}_{i==k}$. All existing exponential integral-based methods set the number of additional timesteps to be the same as the order of Taylor expansion to keep the convergence order (see App. E for details). In our framework, we allow low-order estimation for derivatives as an optional strategy, i.e., the number of additional points could be smaller than Taylor expansion order, since low-order difference is more stable and our experiments in Fig. 2 have revealed the potential of this method. For the scaling operation, we use $1 + R(h)$ to rescale the derivatives. We choose five scaling coefficients $R(h)$ as follows.

- $R_0(h) = 0$ (i.e., no scaling operation),
- $R_1(h) = \frac{\frac{h'}{2}(e^{h'} - 1)}{e^{h'} - h' - 1} - 1$,
- $R_2(h) = \frac{\frac{h'^2}{2}}{e^{h'} - h' - 1} - 1$,
- $R_3(h) = 0.5(e^{h'} - 1)$,
- $R_4(h) = -0.5(e^{h'} - 1)$,

where $h' = h$ when the prediction type is noise prediction, and $h' = -h$ if the prediction is data prediction. The choices of $R_1$ and $R_2$ are inspired by existing methods (Lu et al., 2022a;b; Zhao et al., 2023) (see App. E for details). $R_3$ and $R_4$ are chosen for larger scale coefficients. We demonstrate the $R(h)$-$h$ curve in Fig. 3. All $R(h)$s are small quantities of $\mathcal{O}(h)$ order(except $R_0(h)$) and thus convergence is guaranteed for sufficiently small $h$. Note that though we only try to apply low-order Taylor-difference and scaling operation to the 1-st derivative out of consideration for the potential instability of high-order derivatives and the difficulty in search phase, these methods can be applied to high-order derivatives. Furthermore, in addition to the proposed two-step method with Taylor-difference and scaling, other off-the-shelf numerical differentiation methods can also be applied. These extensions are left for future exploration.

**Corrector.** As discussed in Sec. 3.1, the correcting process can be viewed as an independent step using the function evaluation of the target timestep. Therefore, the searchable strategies listed above can also be applied here, causing a too large search space. To avoid this problem, we simply increase

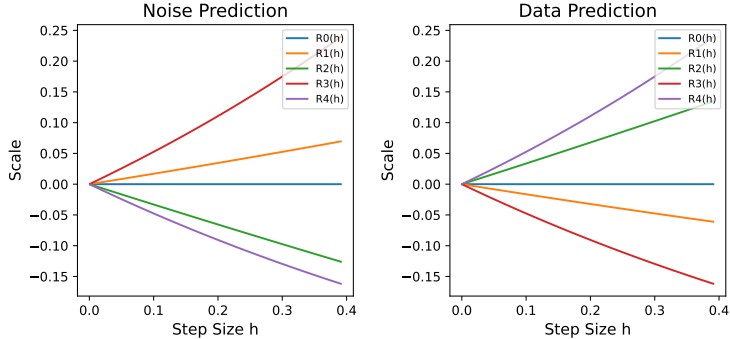

Figure 3: The scales of all $R_h$

Table 4: FID results of independent search of every component in $S^3$ on CIFAR-10. All results are calculated on 50k generated samples.

| NFE | 5 | 7 | 10 |
|---|---|---|---|
| Baseline-B | 23.44 | 6.47 | 3.90 |
| Timestep | **10.06** | **5.36** | **2.88** |
| Order | 17.76 | 5.86 | 3.90 |
| Prediction Type | 23.44 | 6.47 | 3.90 |
| Corrector | 22.58 | 6.23 | 3.90 |
| Scaling | 21.67 | 5.56 | 3.90 |
| Low Order Estimation | 23.44 | 5.91 | 3.84 |

the Taylor expansion order and the Taylor-difference order for all derivatives of the corrector by 1 compared to the corresponding predictor. The prediction type is kept consistent with the predictor. No derivative scaling operations are used for the corrector. In this way, the searchable part only includes whether to use corrector, making the search problem easier to handle. Further exploration of independent strategies for the corrector can be a valuable direction for future research.

## C    ADDITIONAL INVESTIGATE THE POTENTIAL OF EVERY COMPONENT

In this section, we first discuss the improvement potential of each component and offer suggestions on which component should be paid more attention in the future designing of solver schedules. We find from our search results that on most datasets, the timestep schedule and order schedule are the most different with baseline methods among all components. To further validate this observations, we independently search for every single component and fix others. We only evolutionary sample a small amount of solver schedules and truly evaluate them to do the search. The results are listed in Tab. 4 and Tab. 5. From the results we can conclude that *timestep* is the component with the most improvement space for baseline and *order* comes as the second. Moreover, all searchable components can be improved to enhance the sampling quality. We suggest that for future design of solver schedule, the design of *timestep* and *order* schedule should be treated with high priority and the strategies of other components could be tuned for better performance.

## D    ALGORITHM DETAILS

In this section we detail the search process of $S^3$.

### D.1    DETAILED ALGORITHM OF THE UNIFIED SAMPLING FRAMEWORK

We present the detailed general process of our sampling framework $S^3$ in Alg. 1. We mark all searchable components of the framework in *italics*.

Table 5: FID results of independent search of every component in $S^3$ on LSUN-Bedroom. All results are calculated on 1k generated samples.

| NFE | 5 | 7 | 10 |
|---|---|---|---|
| Baseline-B | 33.92 | 25.94 | 23.65 |
| Timestep | **27.23** | **22.99** | **18.64** |
| Order | 31.89 | 25.41 | 22.38 |
| Prediction Type | 32.43 | 24.84 | 23.25 |
| Corrector | 33.40 | 25.60 | 23.27 |
| Scaling | 32.84 | 25.60 | 23.25 |
| Low Order Estimation | 33.92 | 25.62 | 23.52 |

---

**Algorithm 1** Sample through USF

---

1: Get timestep discretization scheme $[t_0, t_1, \cdots, t_N]$.
2: Sample $x_{t_N}$ from $\mathcal{N}(0, \boldsymbol{I})$
3: **for** $i = N - 1, \cdots, 0$ **do**
4:     Get the *prediction type* from **noise prediction** or **data prediction** of neural network.
5:     Get the *starting point* $t_{s_i}$, where $s_i \geq i$.
6:     Get the *order of Taylor expansion* $n_i$, where $n_i \leq N - i$.
7:     $h = \lambda_{t_i} - \lambda_{t_{s_i}}$
8:     **if** noise prediction **then**
9:         $x_{t_i} = \frac{\alpha_{t_i}}{\alpha_{t_{s_i}}} x_{t_{s_i}}$
10:         **for** $k = 0, \cdots, n_i - 1$ **do**
11:             $\tilde{\epsilon}_\theta^{(k)}(x_{t_{s_i}}, t_{s_i}) \leftarrow$ *Estimate* $\epsilon_\theta^{(k)}(x_{t_{s_i}}, t_{s_i})$
12:             $x_{t_i} = x_{t_i} - \sigma_{t_i} h^{k+1} \varphi_{k+1}^\epsilon(h) \tilde{\epsilon}_\theta^{(k)}(x_{t_{s_i}}, t_{s_i})$
13:         **end for**
14:     *Correct* $x_{t_i}$ using $\epsilon_\theta(x_{t_i}, t_i)$
15:     **else if** data prediction **then**
16:         $x_{t_i} = \frac{\sigma_{t_i}}{\sigma_{t_{s_i}}} x_{t_{s_i}}$
17:         **for** $k = 0, \cdots, n_i - 1$ **do**
18:             $\tilde{\epsilon}_\theta^{(k)}(x_{t_{s_i}}, t_{s_i}) \leftarrow$ *Estimate* $\epsilon_\theta^{(k)}(x_{t_{s_i}}, t_{s_i})$
19:             $x_{t_i} = x_{t_i} + \alpha_{t_i} h^{k+1} \varphi_{k+1}^x(h) \tilde{x}_\theta^{(k)}(x_{t_{s_i}}, t_{s_i})$
20:         **end for**
21:     *Correct* $x_{t_i}$ using $x_\theta(x_{t_i}, t_i)$
22:     **end if**
23: **end for**
24: **return** $x_{t_0}$

---

## D.2 DETAILED ALGORITHM OF MULTI-STAGE SEARCH

As discussed in Sec. 4.2, we use a multi-stage search process to search for well-performed solver schedules as described in Alg. 2 and demonstrated in App. D.2. We first initialize the population with baseline solver schedules mentioned in App. F.3 to avoid the 'cold-start' dilemma since most solver schedules in the search space are not good enough. In the first iteration of the search process, we conduct the evolutionary search based on the true performance of all sampled schedules since no predictor can be utilized. At the end of each iteration, we use all evaluated schedules to update the weights of the predictor, which will be used to guide the evolutionary search in the next iteration.

## D.3 PREDICTOR DESIGN

**Architecture**. Our performance predictor P takes parameterized solver schedules s as input and outputs the performance. The predictor contains three modules: 1. timestep encoder $[t_1, \cdots, t_M] \rightarrow [\mathrm{Emb}_1^\mathcal{T}, \cdots, \mathrm{Emb}_M^\mathcal{T}]$, which maps the sequential timesteps to sequential embeddings using a positional embedder followed by a MLP (Vaswani et al., 2017; Ho et al., 2020; Song et al., 2020b);

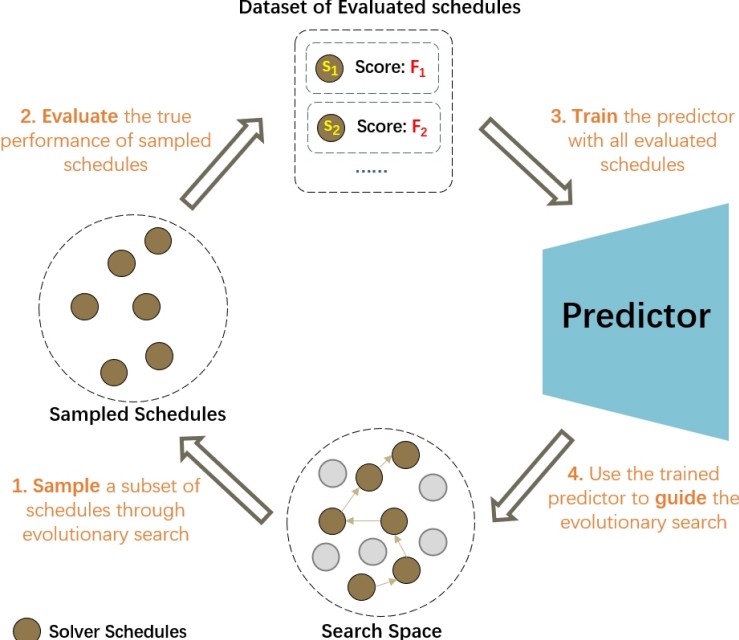

Figure 4: The iterative workflow of predictor-based multi-stage search.

---

**Algorithm 2** Predictor-based Multi-stage Search

---

**Require:**
$\mathcal{S}$: search space of solver schedule
$\mathrm{P} : \mathcal{S} \to \mathbb{R}$: a predictor which takes a parameterized solver schedule as input and outputs its performance
$\mathrm{N} : \mathcal{S} \to \mathbb{N}^+$: get the NFE of a solver schedule

**Hyperparameter:**
$N^{(k)}$: number of solver schedules sampled in the k-th iteration

**Input:**
$\mathbf{C} = [C_1, C_2, \cdots, C_n]$: a series of timestep budgets

**Search Process:**
1: Initialize P randomly
2: Initialize $\tilde{S}$ with baseline solver schedules
3: **for** $k = 1, \cdots, MAX\_ITER\_NUM$ **do**
4:   **if** $k == 1$ **then**
5:     Sample a set of solver schedules $S^{(k)} = \{s_j\}_{j=1,\cdots,N^{(k)}}$ from $\mathcal{S}$ using evolutionary search guided by the ground-truth performance.
6:   **else**
7:     Sample a set of solver schedules $S^{(k)} = \{s_j\}_{j=1,\cdots,N^{(k)}}$ from $\mathcal{S}$ using evolutionary search guided by the predictor P
8:   **end if**
9:   Evaluate the ground-truth performance $p_j$ of all solver schedules $s_j \in S^{(k)}$, get $\tilde{S}^{(k)} = \{(s_j, p_j)\}_{j=1,\cdots,N^{(k)}}$
10:   $\tilde{S} \leftarrow \tilde{S} \cup \tilde{S}^{(k)}$, and $\tilde{S}$ is used to train the predictor P
11: **end for**
12: $s_i^* = \arg\min_{s_j} p_j$, s.t. $\mathrm{N}(s_j) < C_i$
13: **return** $s_i^*$

---

2. encoder of other solving strategies, which uses one-hot embedders followed by MLPs to embed the decisions of one component (e.g., order of Taylor-expansion) $c_k$ and get $[\mathrm{Emb}_1^k, \cdots, \mathrm{Emb}_M^k]$. Embeddings of decisions from all components are concatenated to obtain the final embedding of solving strategies $[\mathrm{Emb}_1^c, \cdots, \mathrm{Emb}_M^c] = [\mathrm{Emb}_1^1|\cdots|\mathrm{Emb}_1^K, \cdots, \mathrm{Emb}_M^1|\cdots|\mathrm{Emb}_M^K]$, where $K$ is the number of searchable components in the our framework of solving Eq. (6); 3. sequence pre-

dictor, which takes the concatenated embedding $[\text{Emb}_1^{\mathcal{T}}|\text{Emb}_1^k, \cdots, \text{Emb}_M^{\mathcal{T}}|\text{Emb}_M^k]$ as input and outputs the final predicted score. An LSTM (Merity et al., 2017) is used to process the sequential embedding, followed by an MLP to regress the final score.

**Training**. Since ranking information of different schedules is far more important than their absolute metrics, we use pair-wise ranking loss to train the predictor, which is validated to be more effective in extracting relative quality information and preserving the ranking information (Ning et al., 2020; Liu et al., 2023).

$$loss = \sum_{i=1}^{\|\tilde{S}\|} \sum_{j, \mathcal{F}(s_j) > \mathcal{F}(s_i)} \max(0, m - (\text{P}(s_j) - \text{P}(s_i))), \tag{11}$$

where $\tilde{S}$ is the dataset of schedule-performance pair data. Lower output value of the predictor trained with ranking loss indicates higher performance of the input solver schedule.

# E  RELATIONSHIP BETWEEN USF AND EXISTING EXPONENTIAL INTEGRAL BASED METHODS

In this section, we discuss the relationship between existing exponential integral methods (Zhang & Chen, 2022; Lu et al., 2022a;b; Zhao et al., 2023) and USF. We show that all existing methods in the $\lambda$ domain (Lu et al., 2022a;b; Zhao et al., 2023) are included in our framework by assigning corresponding decisions to every component. The method in $t$ domain (Zhang & Chen, 2022) also shares strong relevance with USF.

## E.1  DPM-SOLVER

DPM-Solver (Lu et al., 2022a) is a singlestep solver based on the noise prediction type Taylor expansion of exponential integral. Lu et al. (2022a) gives two examples of DPM-Solver-2 and DPM-Solver-3 in Algorithm-1 and Algorithm-2 of its paper, correspondingly. We discuss the relationship between these two examples and USF below.

**DPM-Solver-2.** In DPM-Solver-2, to calculate the trajectory value $\tilde{x}_{t_i}$ of the target point $t_i$ from the starting point $t_{i-1}$, a midpoint $s_i \in (t_i, t_{i-1})$ is given by $s_i = t_\lambda(\frac{\lambda_{t_{i-1}} + \lambda_{t_i}}{2})$ and its trajectory value $u_i$ is calculated through the 1-st expansion of Eq. (8) (i.e., DDIM (Song et al., 2020a)) from the starting point $t_{i-1}$. Then $\tilde{x}_{t_i}$ is given by: $\tilde{x}_{t_i} = \frac{\alpha_{t_i}}{\alpha_{t_{i-1}}} \tilde{x}_{t_{i-1}} - \sigma_{t_i}(e^{h_i} - 1)\epsilon_\theta(u_i, s_i)$ (see the Algorithm-1 in the original paper Lu et al. (2022a) for details). We further write this formula:

$$
\begin{aligned}
\tilde{x}_{t_i} =& \frac{\alpha_{t_i}}{\alpha_{t_{i-1}}} \tilde{x}_{t_{i-1}} - \sigma_{t_i}(e^{h_i} - 1)\epsilon_\theta(u_i, s_i) \\
=& \frac{\alpha_{t_i}}{\alpha_{t_{i-1}}} \tilde{x}_{ti-1} - \sigma_{t_i}(e^{h_i} - 1)\epsilon_\theta(\tilde{x}_{t_{i-1}}, t_{i-1}) - \sigma_{t_i}(e^{h_i} - 1)(\epsilon_\theta(u_i, s_i) - \epsilon_\theta(\tilde{x}_{t_{i-1}}, t_{i-1})) \\
=& \frac{\alpha_{t_i}}{\alpha_{t_{i-1}}} \tilde{x}_{t_{i-1}} - \sigma_{t_i}(e^{h_i} - 1)\epsilon_\theta(\tilde{x}_{t_{i-1}}, t_{i-1}) \\
& - \sigma_{t_i} \frac{(e^{h_i} - 1)(\lambda_{s_i} - \lambda_{t_{i-1}})}{e^{h_i} - h_i - 1} h_i^2 \varphi_2^\epsilon(h_i) \frac{\epsilon_\theta(u_i, s_i) - \epsilon_\theta(\tilde{x}_{t_{i-1}}, t_{i-1})}{\lambda_{s_i} - \lambda_{t_{i-1}}} \\
=& \frac{\alpha_{t_i}}{\alpha_{t_{i-1}}} \tilde{x}_{t_{i-1}} - \sigma_{t_i}(e^{h_i} - 1)\epsilon_\theta(\tilde{x}_{t_{i-1}}, t_{i-1}) - \sigma_{t_i} h_i^2 \varphi_2^\epsilon(h_i)(1 + R_1(h_i)) \frac{\epsilon_\theta(u_i, s_i) - \epsilon_\theta(\tilde{x}_{t_{i-1}}, t_{i-1})}{\lambda_{s_i} - \lambda_{t_{i-1}}}.
\end{aligned}
$$

We can see that the 2-nd solver expands Eq. (8) to the 2-nd order and uses $1 + R_1(h_i)$ to scale the 1-st derivative estimated by Taylor-difference (degrade to direct two-points difference approximation in this case). Therefore, DPM-Solver-2 can be viewed as two updates with Taylor expansion orders 1 and 2. The second update uses second point $t_{i-1}$ before the target timestep $t_i$ as the starting point and $1 + R_1(h)$ to scale the 1-st derivative.

**DPM-Solver-3.** In DPM-Solver-3, two midpoints $s_{2i-1}$ and $s_{2i}$ are selected to calculate the final solution $x_{t_i}$. Firstly, the trajectory value of $s_{2i-1}$ is calculated by the 1-st solver DDIM (Song et al., 2020a). Then, the trajectory value of $s_{2i}$ is calculated from the starting point $t_{i-1}$ and the previous

midpoint $s_{2i-1}$.

$$
\begin{aligned}
u_{2i} =& \frac{\alpha_{s_{2i}}}{\alpha_{t_{i-1}}} \tilde{x}_{t_{i-1}} - \sigma_{s_{2i}}(e^{\lambda_{s_{2i}}-\lambda_{t_{i-1}}} - 1)\epsilon_\theta(\tilde{x}_{t_{i-1}}, t_{i-1}) \\
& - \frac{\sigma_{s_{2i}}(\lambda_{s_{2i}} - \lambda_{i_{i-1}})}{\lambda_{s_{2i-1}} - \lambda_{i_{i-1}}} (\frac{e^{\lambda_{s_{2i}}-\lambda_{t_{i-1}}} - 1}{\lambda_{s_{2i}} - \lambda_{t_{i-1}}} - 1)(\epsilon_\theta(u_{2i-1}, s_{2i-1}) - \epsilon_\theta(\tilde{x}_{t_{i-1}}, t_{i-1})) \\
=& \frac{\alpha_{s_{2i}}}{\alpha_{t_{i-1}}} \tilde{x}_{t_{i-1}} - \sigma_{s_{2i}}(e^{\lambda_{s_{2i}}-\lambda_{t_{i-1}}} - 1)\epsilon_\theta(\tilde{x}_{t_{i-1}}, t_{i-1}) \\
& - \sigma_{s_{2i}}(\lambda_{s_{2i}} - \lambda_{t_{i-1}})^2 \varphi_2^\epsilon(\lambda_{s_{2i}} - \lambda_{t_{i-1}}) \frac{\epsilon_\theta(u_{2i-1}, s_{2i-1}) - \epsilon_\theta(\tilde{x}_{i-1}, t_{i-1})}{\lambda_{s_{2i}} - \lambda_{t_{i-1}}}.
\end{aligned}
$$

This process also uses the 2-nd order expansion of Eq. (8) like DPM-Solver-2, but uses $1 + R_0(\lambda_{s_{2i}} - \lambda_{t_{i-1}})$ rather than $1 + R_1$ in DPM-Solver-2. The final calculation of $\tilde{x}_{t_i}$ is similar to the above process but uses $s_{2i}$ as the midpoint of this 2-nd singlestep solver. In conclusion, DPM-Solver-3 can be viewed as three updates with Taylor expansion orders 1, 2, and 2. The last two solvers both use $t_{i-1}$ as the starting point.

### E.2 DPM-SOLVER++

DPM-Solver++ (Lu et al., 2022b) propose several new solving strategies based on the framework of DPM-Solver (Lu et al., 2022a). The first is to switch the noise prediction model to the data prediction model, both of which are optional strategies in USF. The second is to apply multistep solvers. The original paper provides the process of DPM-Solver++(2S) and DPM-Solver++(2M) in Algorithm 1 and Algorithm 2. We discuss the relationship between these two examples and USF as follows.

**DPM-Solver++(2S).** DPM-Solver++(2S) is similarly designed to DPM-Solver-2 Lu et al. (2022a), except for switch to data prediction. See the Algorithm 1 in Lu et al. (2022b) for details. We can draw an analogous conclusion to DPM-Solver-2 by rewriting the formula of the 2-nd update.

$$
\begin{aligned}
\tilde{x}_{t_i} =& \frac{\sigma_{t_i}}{\sigma_{t_{i-1}}} \tilde{x}_{t_{i-1}} - \alpha_{t_i}(e^{-h_i} - 1)x_\theta(\tilde{x}_{t_{i-1}}, t_{i-1}) - \alpha_{t_i} \frac{(e^{-h_i} - 1)h_i}{2} \frac{x_\theta(\tilde{x}_{s_i}, s_i) - x_\theta(\tilde{x}_{t_{i-1}}, t_{i-1})}{\lambda_{s_i} - \lambda_{t_{i-1}}} \\
=& \frac{\sigma_{t_i}}{\sigma_{t_{i-1}}} \tilde{x}_{t_{i-1}} - \alpha_{t_i}(e^{-h_i} - 1)x_\theta(\tilde{x}_{t_{i-1}}, t_{i-1}) \\
& - \alpha_{t_i} \frac{\frac{h_i}{2}(e^{-h_i} - 1)}{e^{-h_i} - 1 + h_i} h_i^2 \varphi_2^x(h_i) \frac{x_\theta(\tilde{x}_{s_i}, s_i) - x_\theta(\tilde{x}_{t_{i-1}}, t_{i-1})}{\lambda_{s_i} - \lambda_{t_{i-1}}} \\
=& \frac{\sigma_{t_i}}{\sigma_{t_{i-1}}} \tilde{x}_{t_{i-1}} - \alpha_{t_i}(e^{-h_i} - 1)x_\theta(\tilde{x}_{t_{i-1}}, t_{i-1}) \\
& - \alpha_{t_i}(1 + R_1(h_i))h_i^2 \varphi_2^x(h_i) \frac{x_\theta(\tilde{x}_{s_i}, s_i) - x_\theta(\tilde{x}_{t_{i-1}}, t_{i-1})}{\lambda_{s_i} - \lambda_{t_{i-1}}}
\end{aligned}
$$

We can see that DPM-Solver++(2S) also uses $R_1(h)$ to scale the 1-st derivative. Therefore, prediction type is the only difference between the two methods.

**DPM-Solver++(2M).** DPM-Solver++(2M) no longer uses singlestep update. Despite the first step, which needs a cold start with the 1-st solver, the rest of the steps use 2-nd expansion of Eq. (9) (See

the Algorithm 2 in Lu et al. (2022b) for details). We reformulate the 2-nd update formula as below.

$$
\begin{aligned}
\tilde{x}_{t_i} =\ & \frac{\sigma_{t_i}}{\sigma_{t_{i-1}}}\tilde{x}_{t_{i-1}} - \alpha_{t_i}(e^{-h_i}-1)x_\theta(\tilde{x}_{t_{i-1}},t_{i-1}) - \alpha_{t_i}\frac{(e^{-h_i}-1)h_i}{2}\frac{x_\theta(\tilde{x}_{t_{i-1}},t_{i-1})-x_\theta(\tilde{x}_{t_{i-2}},t_{i-2})}{\lambda_{t_{i-1}}-\lambda_{t_{i-2}}} \\
=\ & \frac{\sigma_{t_i}}{\sigma_{t_{i-1}}}\tilde{x}_{t_{i-1}} - \alpha_{t_i}(e^{-h_i}-1)x_\theta(\tilde{x}_{t_{i-1}},t_{i-1}) \\
& - \alpha_{t_i}\frac{\frac{h_i}{2}(e^{-h_i}-1)}{e^{-h_i}-1+h_i}h_i^2\varphi_2^x(h_i)\frac{x_\theta(\tilde{x}_{t_{i-1}},t_{i-1})-x_\theta(\tilde{x}_{t_{i-2}},t_{i-2})}{\lambda_{t_{i-1}}-\lambda_{t_{i-2}}} \\
=\ & \frac{\sigma_{t_i}}{\sigma_{t_{i-1}}}\tilde{x}_{t_{i-1}} - \alpha_{t_i}(e^{-h_i}-1)x_\theta(\tilde{x}_{t_{i-1}},t_{i-1}) \\
& - \alpha_{t_i}(1+R_1(h_i))h_i^2\varphi_2^x(h_i)\frac{x_\theta(\tilde{x}_{t_{i-1}},t_{i-1})-x_\theta(\tilde{x}_{t_{i-2}},t_{i-2})}{\lambda_{t_{i-1}}-\lambda_{t_{i-2}}}
\end{aligned}
$$

We find that the only difference between DPM-Solver++(2M) and the 2-nd update of DPM-Solver++(2S) is the position of starting point. In the 2-nd update of DPM-Solver++(2S), the starting point is the second point before the target point. In DPM-Solver++(2M), the starting point is the first point before the target point.

### E.3 UNIPC

UniPC (Zhao et al., 2023) proposes two update methods, UniP and UniC. In fact, UniC can be viewed as a special version of UniP with the involvement of the function value $\epsilon_\theta(x_t,t)$ or $x_\theta(x_t,t)$ at the target point $t$. Therefore, we mainly discuss the relationship between UniP and USF.

**UniP**. Take noise prediction model as an example, the update formula of Uni-P is given by $\tilde{x}_{t_i} = \frac{\alpha_{t_i}}{\alpha_{t_{i-1}}}\tilde{x}_{t_{i-1}} - \sigma_{t_i}(e^{h_i}-1)\epsilon_\theta(\tilde{x}_{t_{i-1}},t_{i-1}) - \sigma_{t_i}B(h_i)\sum_{m=1}^{p-1}\frac{a_m D_m}{r_m}$, where $D_m = \epsilon_\theta(x_{t_{i-1}},t_{i-1}) - \epsilon_\theta(x_{t_{i_m}},t_{i_m})$, $\boldsymbol{a_{p-1}} = [a_1,a_2,\cdots,a_{p-1}]$ satisfies $\boldsymbol{a_{p-1}} = B^{-1}(h_i)\boldsymbol{R}_{p-1}^{-1}(h_i)\Phi_{p-1}(h_i)$, and $\boldsymbol{R}_{p-1}(h)$ and $\Phi_{p-1}(h)$ are given below (see the Algorithm 2 in Zhao et al. (2023)).

$$
\boldsymbol{R}_{p-1}(h) = \begin{bmatrix} 1 & 1 & \cdots & 1 \\ r_1 h & r_2 h & \cdots & r_{p-1}h \\ \cdots & \cdots & \cdots & \cdots \\ (r_1 h)^{p-2} & (r_2 h)^{p-2} & \cdots & (r_{p-1}h)^{p-2} \end{bmatrix},
$$

$$
\Phi_{p-1}(h) = [1!h\varphi_2^\epsilon(h), 2!h^2\varphi_3^\epsilon(h), \cdots, (p-1)!h^{p-1}\varphi_p^\epsilon(h)].
$$

Then suppose the solution given by our USF with $p$-th Taylor expansion and pure Taylor-difference estimation for each derivative is $\tilde{x}_{t_i}^U$. We state that $\tilde{x}_{t_i}^U = \tilde{x}_{t_i}$ if the starting point and all additional points in the two solvers are the same. The proof is as follows.

*Proof.*

$$
\tilde{x}_{t_i}^U = \frac{\alpha_{t_i}}{\alpha_{t_{i-1}}}x_{t_{i-1}} - \sigma_{t_i}\sum_{k=0}^{p-1}h^{k+1}\varphi_{k+1}^\epsilon(h)\tilde{\epsilon}_\theta^{(k)}(x_{t_{i-1}},t_{i-1}) \tag{12}
$$

$$
= \frac{\alpha_{t_i}}{\alpha_{t_{i-1}}}x_{t_{i-1}} - \sigma_{t_i}(e^{h_i}-1)\epsilon_\theta(\tilde{x}_{t_{i-1}},t_{i-1}) - \sigma_{t_i}\sum_{k=1}^{p-1}h^{k+1}\varphi_{k+1}^\epsilon(h)\boldsymbol{D}_U^T\boldsymbol{a}_U/h_i^k, \tag{13}
$$

where $\boldsymbol{D}_U = [\epsilon_\theta(x_{t_{i-1}},t_{i-1})-\epsilon_\theta(x_{t_{i_1}},t_{i_1}),\cdots,\epsilon_\theta(x_{t_{i-1}},t_{i-1})-\epsilon_\theta(x_{t_{i_{p-1}}},t_{i_{p-1}})]$, and $\boldsymbol{a}_U$ satisfies $\boldsymbol{a}_U = \boldsymbol{C}_U^{-1}\boldsymbol{b}_U^k$. As discussed in Def. B.1, $\boldsymbol{C}_U$ and $\boldsymbol{b}_U^k$ are written as:

$$
\boldsymbol{C}_U = \begin{bmatrix} r_1 & r_2 & \cdots & r_{p-1} \\ r_1^2/2! & r_2^2/2! & \cdots & r_{p-1}^2/2! \\ \cdots & \cdots & \cdots & \cdots \\ r_1^{p-1}/(p-1)! & r_2^{p-1}/(p-1)! & \cdots & r_{p-1}^{p-1}/(p-1))! \end{bmatrix},
$$

$$
\boldsymbol{b}_U^k = [b_1, b_2, \cdots, b_{p-1}], b_i = \mathbb{I}_{i==k}
$$

substitute $\boldsymbol{D}_U$, $\boldsymbol{C}_U$ and $\boldsymbol{b}_U^k$ to Eq. (13):

$$\tilde{x}_{t_i}^U = \frac{\alpha_{t_i}}{\alpha_{t_{i-1}}} x_{t_{i-1}} - \sigma_{t_i}(e^{h_i} - 1)\epsilon_\theta(\tilde{x}_{t_{i-1}}, t_{i-1}) - \sigma_{t_i} \sum_{k=1}^{p-1} h\varphi_{k+1}^\epsilon(h)\boldsymbol{D}_U^T\boldsymbol{C}_U^{-1}\boldsymbol{b}_U^k \tag{14}$$

$$= \frac{\alpha_{t_i}}{\alpha_{t_{i-1}}} x_{t_{i-1}} - \sigma_{t_i}(e^{h_i} - 1)\epsilon_\theta(\tilde{x}_{t_{i-1}}, t_{i-1}) - \sigma_{t_i} h\boldsymbol{D}_U^T \sum_{k=1}^{p-1} \varphi_{k+1}^\epsilon(h)\boldsymbol{C}_U^{-1}\boldsymbol{b}_U^k \tag{15}$$

$$= \frac{\alpha_{t_i}}{\alpha_{t_{i-1}}} x_{t_{i-1}} - \sigma_{t_i}(e^{h_i} - 1)\epsilon_\theta(\tilde{x}_{t_{i-1}}, t_{i-1}) - \sigma_{t_i} \sum_{m=1}^{p-1} D_m q_m, \tag{16}$$

where $q_m$ is the $m-th$ element of vector $\boldsymbol{q} = h\sum_{k=1}^{p-1}\varphi_{k+1}^\epsilon(h)\boldsymbol{C}_U^{-1}\boldsymbol{b}_U^k = \boldsymbol{C}_U^{-1}\boldsymbol{b}_U'$, where $\boldsymbol{b}_U' = [h\varphi_2^\epsilon(h), h\varphi_3^\epsilon(h), \cdots, h\varphi_p^\epsilon(h)]$. Then we rewrite this equation by left multiply $\boldsymbol{C}_U$ at both sides:

$$\boldsymbol{C}_U\boldsymbol{q} = \begin{bmatrix} r_1 & r_2 & \cdots & r_{p-1} \\ r_1^2/2! & r_2^2/2! & \cdots & r_{p-1}^2/2! \\ \cdots & \cdots & \cdots & \cdots \\ r_1^{p-1}/(p-1)! & r_2^{p-1}/(p-1)! & \cdots & r_{p-1}^{p-1}/(p-1))! \end{bmatrix} \begin{bmatrix} q_1 \\ q_2 \\ \cdots \\ q_{p-1} \end{bmatrix} \tag{17}$$

$$= \boldsymbol{b}_U' = \begin{bmatrix} h\varphi_2^\epsilon(h) \\ h\varphi_3^\epsilon(h) \\ \cdots \\ h\varphi_p^\epsilon(h) \end{bmatrix} \tag{18}$$

This equation is equivalent to:

$$\begin{bmatrix} 1 & 1 & \cdots & 1 \\ r_1 h & r_2 h & \cdots & r_{p-1} h \\ \cdots & \cdots & \cdots & \cdots \\ (r_1 h)^{p-2} & (r_2 h)^{p-2} & \cdots & (r_{p-1} h)^{p-2} \end{bmatrix} \begin{bmatrix} r_1 q_1 \\ r_2 q_2 \\ \cdots \\ r_{p-1} q_{p-1} \end{bmatrix} = \begin{bmatrix} h\varphi_2^\epsilon(h) \\ 2!h^2\varphi_3^\epsilon(h) \\ \cdots \\ (p-1)!h^{p-1}\varphi_p^\epsilon(h) \end{bmatrix} \tag{19}$$

Compare Eq. (19) with $\boldsymbol{a_{p-1}} = B^{-1}(h_i)\boldsymbol{R}_{p-1}^{-1}(h_i)\Phi_{p-1}(h_i)$, it is obvious to obtain that $r_m q_m = a_m B(h_i)$. Substitute this equation to Eq. (16), we get:

$$\tilde{x}_{t_i}^U = \frac{\alpha_{t_i}}{\alpha_{t_{i-1}}} x_{t_{i-1}} - \sigma_{t_i}(e^{h_i} - 1)\epsilon_\theta(\tilde{x}_{t_{i-1}}, t_{i-1}) - \sigma_{t_i} B(h_i) \sum_{m=1}^{p-1} \frac{a_m D_m}{r_m} = \tilde{x}_{t_i} \tag{20}$$

$\square$

We can see that UniP is equivalent to using full-order Taylor-difference to estimate all derivatives. The coefficient $B(h)$ is eliminated here and thus has no impact on the sample quality. However, the implementation (see https://github.com/wl-zhao/UniPC for details) of UniP applies a special case for the 2-nd solver:

$$\tilde{x}_{t_i} = \frac{\alpha_{t_i}}{\alpha_{t_{i-1}}} x_{t_{i-1}} - \sigma_{t_i}(e^{h_i} - 1)\epsilon_\theta(\tilde{x}_{t_{i-1}}, t_{i-1}) - \sigma_{t_i}\frac{B(h_i)}{2}\frac{D_1}{r_1}$$

$$= \frac{\alpha_{t_i}}{\alpha_{t_{i-1}}} x_{t_{i-1}} - \sigma_{t_i}(e^{h_i} - 1)\epsilon_\theta(\tilde{x}_{t_{i-1}}, t_{i-1})$$

$$- \sigma_{t_i}\frac{h_i B(h_i)}{2(e^{h_i} - h_i - 1)}h_i^2\varphi_2^\epsilon(h_i)\frac{\epsilon_\theta(x_{t_{i-1}}, t_{i-1}) - \epsilon_\theta(x_{t_{i_1}}, t_{i_1})}{\lambda_{t_{i-1}} - \lambda_{t_{i_1}}}$$

This is equivalent to scaling 1-st derivative with $1 + R(h) = \frac{h_i B(h_i)}{2(e^{h_i} - h_i - 1)}$. UniPC (Zhao et al., 2023) empirically provides two choices for $B(h)$: $B_1(h) = h$ and $B_2(h) = e^h - 1$. The corresponding $R(h)$ for $B_1(h)$ is $\frac{\frac{h^2}{2}}{e^h - h - 1} - 1 = R_2(h)$ and the corresponding $R(h)$ for $B_2(h)$ is $\frac{\frac{h}{2}(e^h - 1)}{e^h - h - 1} - 1 = R_1(h)$.

In conclusion, 2-nd order UniP is equivalent to using $R_1$ or $R_2$ to scale the 1-st derivative, while UniP with other orders is equivalent to applying full-order Taylor-difference method to all derivatives without scaling.

**UniC.** UniC has no fundamental difference with UniP. Inspired by it, our framework USF set the corrector to be a searchable component without re-evaluation of the function at the target timestep.

### E.4 DEIS

As discussed in Sec. 2.2.2, DEIS uses Lagrange interpolation to calculate the integral in $t$ **domain** directly without Taylor expansion, as given below.

$$\tilde{x}_{t_{i-1}} = \frac{\alpha_{t_{i-1}}}{\alpha_{t_i}} x_{t_i} - \alpha_{t_{i-1}} \int_{t_i}^{t_{i-1}} e^{-\lambda(\tau)} \lambda'(\tau) \tilde{\epsilon}_\theta(x_\tau, \tau) \mathrm{d}\tau \tag{21}$$

$$= \frac{\alpha_{t_{i-1}}}{\alpha_{t_i}} x_{t_i} - \alpha_{t_{i-1}} \int_{t_i}^{t_{i-1}} e^{-\lambda(\tau)} \lambda'(\tau) \sum_{j=0}^{p} (\prod_{k \neq j}^{p} \frac{\tau - t_{i+k}}{t_{i+j} - t_{i+k}}) \epsilon_\theta(x_{t_{i+j}}, t_{i+j}) \mathrm{d}\tau, \tag{22}$$

Note that other methods (Lu et al., 2022a;b; Zhao et al., 2023) including USF estimate the integral in the $\lambda$ domain, so DEIS has fundamental differences with these methods. However, we show that the numerical integral estimation method of DEIS (i.e., Lagrange interpolation), which is one of the core designs of this work, can still be incorporated in USF in the $\lambda$ domain. We rewrite Eq. (22) w.r.t. $\lambda$:

$$\tilde{x}_{t_{i-1}} = \frac{\alpha_{t_{i-1}}}{\alpha_{t_i}} x_{t_i} - \alpha_{t_{i-1}} \int_{\lambda_{t_i}}^{\lambda_{t_{i-1}}} e^{-\lambda} \tilde{\epsilon}_\theta(x_\tau, \tau) \mathrm{d}\lambda \tag{23}$$

$$= \frac{\alpha_{t_{i-1}}}{\alpha_{t_i}} x_{t_i} - \alpha_{t_{i-1}} \int_{\lambda_{t_i}}^{\lambda_{t_{i-1}}} e^{-\lambda} \sum_{j=0}^{p} (\prod_{k \neq j}^{p} \frac{\lambda - \lambda_{t_{i+k}}}{\lambda_{t_{i+j}} - \lambda_{t_{i+k}}}) \epsilon_\theta(x_{t_{i+j}}, \lambda_{t_{i+j}}) \mathrm{d}\lambda. \tag{24}$$

Lagrange interpolation with $p+1$ points $\epsilon_\theta(x_{t_{i+j}}, \lambda_{t_{i+j}})$ $(j = 0, \cdots, p)$ constructs a polynomial $\tilde{\epsilon}_\theta(x_t, \lambda_t) = a_p \lambda^p + a_{p-1} \lambda^{p-1} + \cdots + a_1 \lambda + a_0$, which pass through all points. We rewrite this polynomial to make the starting point $\lambda_{t_i}$ be the 'center': $\tilde{\epsilon}_\theta(x_t, \lambda_t) = \frac{a_p}{p!}(\lambda - \lambda_{t_i})^p + \frac{a_{p-1}}{(p-1)!}(\lambda - \lambda_{t_i})^{p-1} + \cdots + a_1(\lambda - \lambda_{t_i}) + \epsilon_\theta(x_{t_i}, \lambda_{t_i})$, and construct a linear equation system to solve all coefficients.

$$\begin{bmatrix} \lambda_{t_{i+1}} - \lambda_{t_i} & (\lambda_{t_{i+1}} - \lambda_{t_i})^2/2! & \cdots & (\lambda_{t_{i+1}} - \lambda_{t_i})^p/p! \\ \lambda_{t_{i+2}} - \lambda_{t_i} & (\lambda_{t_{i+2}} - \lambda_{t_i})^2/2! & \cdots & (\lambda_{t_{i+2}} - \lambda_{t_i})^p/p! \\ \cdots & \cdots & \cdots & \cdots \\ \lambda_{t_{i+p}} - \lambda_{t_i} & (\lambda_{t_{i+p}} - \lambda_{t_i})^2/2! & \cdots & (\lambda_{t_{i+p}} - \lambda_{t_i})^p/p! \end{bmatrix} \begin{bmatrix} a_1 \\ a_2 \\ \cdots \\ a_p \end{bmatrix}$$

$$= \begin{bmatrix} \epsilon_\theta(x_{t_{i+1}}, \lambda_{t_{i+1}}) - \epsilon_\theta(x_{t_i}, \lambda_{t_i}) \\ \epsilon_\theta(x_{t_{i+2}}, \lambda_{t_{i+2}}) - \epsilon_\theta(x_{t_i}, \lambda_{t_i}) \\ \cdots \\ \epsilon_\theta(x_{t_{i+p}}, \lambda_{t_{i+p}}) - \epsilon_\theta(x_{t_i}, \lambda_{t_i}) \end{bmatrix}$$

Suppose $h_i = \lambda_{t_{i-1}} - \lambda_{t_i}$, $\lambda_{t_{i+k}} - \lambda_{t_i} = r_k h_i$, and $\epsilon_\theta(x_{t_{i+k}}, \lambda_{t_{i+k}}) - \epsilon_\theta(x_{t_i}, \lambda_{t_i}) = D_k$. Then rewrite the above equation:

$$\begin{bmatrix} a_1 & a_2 & \cdots & a_p \end{bmatrix} \begin{bmatrix} r_1 & r_2 & \cdots & r_{p-1} \\ r_1^2/2! & r_2^2/2! & \cdots & r_{p-1}^2/2! \\ \cdots & \cdots & \cdots & \cdots \\ r_1^{p-1}/(p-1)! & r_2^{p-1}/(p-1)! & \cdots & r_{p-1}^{p-1}/(p-1))! \end{bmatrix}$$

$$= \begin{bmatrix} D_1 & D_2 & \cdots & D_p \end{bmatrix}$$

Extract the $k$-th column of the both sides.

$$a_k = \begin{bmatrix} D_1 & D_2 & \cdots & D_p \end{bmatrix} \begin{bmatrix} r_1 & r_2 & \cdots & r_{p-1} \\ r_1^2/2! & r_2^2/2! & \cdots & r_{p-1}^2/2! \\ \cdots & \cdots & \cdots & \cdots \\ r_1^{p-1}/(p-1)! & r_2^{p-1}/(p-1)! & \cdots & r_{p-1}^{p-1}/(p-1))! \end{bmatrix}^{-1} \begin{bmatrix} 0 \\ 0 \\ \cdots \\ 1 \\ \cdots \\ 0 \end{bmatrix}$$

$$= \boldsymbol{D}\boldsymbol{C}^{-1}\boldsymbol{b}$$

The value of $\boldsymbol{D}$, $\boldsymbol{C}$, and $\boldsymbol{b}$ can be found in Def. B.1 by replacing $f$ with $\epsilon_\theta$ and $t$ with $\lambda$. We can see that $a_k$ is equal to the $k$-th derivative at timestep $t_i$ calculated by full order Taylor-difference. Therefore, the update formula using USF with an expansion order $p+1$ and full order Taylor-difference estimation for all derivatives is given by:

$$\tilde{x}^U_{t_{i-1}} = \frac{\alpha_{t_{i-1}}}{\alpha_{t_i}} x_{t_i} - \sigma_{t_{i-1}} \sum_{k=0}^{p} h_i^{k+1} \varphi^\epsilon_{k+1}(h_i) a_k.$$

And the update formula of DEIS in $\lambda$ domain is written as:

$$\tilde{x}_{t_{i-1}} = \frac{\alpha_{t_{i-1}}}{\alpha_{t_i}} x_{t_i} - \alpha_{t_{i-1}} \int_{\lambda_{t_i}}^{\lambda_{t_{i-1}}} e^{-\lambda} \sum_{k=1}^{p} a^k (\lambda - \lambda_{t_i})^k \mathrm{d}\lambda.$$

It is proved in App. H that $\sigma_{t_{i-1}} h_i^{k+1} \phi^\epsilon_{k+1}(h_i) = \alpha_{t_{i-1}} \int_{\lambda_{t_i}}^{\lambda_{t_{i-1}}} e^{-\lambda} (\lambda - \lambda_{t_i})^k \mathrm{d}\lambda$, so that $\tilde{x}^U_{t_{i-1}} = \tilde{x}_{t_{i-1}}$. In summary, estimating the integral through Lagrange interpolation in $\lambda$ domain is equivalent to using full order Taylor-difference to estimate all derivatives. Thus, DEIS in $\lambda$ domain can also be incorporated in USF.

## F  EXPERIMENTAL SETTINGS

In this section, we list the settings of our experiments.

### F.1  MODELS

All models we use in the experiments are open-source pre-trained models. We list the information of them below.

- **CIFAR-10** We use the model *cifar10_ddpmpp_deep_continuous_steps* provided by the paper Song et al. (2020b). This model applies linear noise schedule with VP-SDE and is trained with continuous-time. It can be got by downloading the *checkpoint_8.pth* at `https://drive.google.com/drive/folders/1ZMLBiu9j7-rpdTQu8M2LlHAEQq4xRYrj`.

- **CelebA** We use the model provided by the paper Song et al. (2020a). This model applies linear noise schedule with VP-SDE and is trained with discrete-time. It can be downloaded at `https://drive.google.com/file/d/1R_H-fJYXSH79wfSKs9D-fuKQVan5L-GR/view`.

- **ImageNet-64** We use the unconditional model trained with L_hybrid provided by the paper Nichol & Dhariwal (2021). This model applies cosine noise schedule with VP-SDE and is trained with discrete-time. It can be downloaded at `https://openaipublic.blob.core.windows.net/diffusion/march-2021/imagenet64_uncond_100M_1500K.pt`.

- **LSUN-Bedroom** We use the model provided by the paper Dhariwal & Nichol (2021). This model applies linear noise schedule with VP-SDE and is trained with discrete-time. It can be downloaded at `https://openaipublic.blob.core.windows.net/diffusion/jul-2021/lsun_bedroom.pt`.

- **ImageNet-128 (classifier guidance)** We use the model provided by the paper Dhariwal & Nichol (2021). This model applies linear noise schedule with VP-SDE and is trained with discrete-time. It can be downloaded at `https://openaipublic.blob.core.windows.net/diffusion/jul-2021/128x128_diffusion.pt`. The classifier is also provided by this paper at `https://openaipublic.blob.core.windows.net/diffusion/jul-2021/128x128_classifier.pt`

- **ImageNet-256 (classifier guidance)** We use the model provided by the paper Dhariwal & Nichol (2021). This model applies linear noise schedule with VP-SDE and is trained with discrete-time. It can be downloaded at `https://openaipublic.blob.core.windows.net/diffusion/jul-2021/256x256_diffusion.pt`. The classifier is also provided by this paper at `https://openaipublic.blob.core.windows.net/diffusion/jul-2021/256x256_classifier.pt`.

- **MS-COCO 256×256 (classifier-free guidance (Ho & Salimans, 2022))** We use the model provided by the paper Rombach et al. (2022). This model applies linear noise schedule with VP-SDE and is trained with discrete-time. Additionally, this model is trained in the latent space. It can be downloaded at `https://huggingface.co/runwayml/stable-diffusion-v1-5`.

## F.2 Evaluation Settings

We sample 1k images to evaluate the FID as a proxy to the final performance in the search phase. For the final evaluation, we sample 50k images for evaluation on all unconditional datasets and ImageNet-128 (classifier guidance) dataset and 10k images on ImageNet-256 (classifier guidance) and MS-COCO 256×256. For CIFAR-10 and CelebA, we get the target statistics by calculating the activations of all images in the training set using the *pytorch-fid* repository at `https://github.com/mseitzer/pytorch-fid`. For ImageNet-64, ImageNet-128, ImageNet-256, and LSUN-Bedroom, we get the target statistics at `https://github.com/openai/guided-diffusion/tree/main/evaluations`. For MS-COCO 256×256, we calculate the activations of all corresponding images with their caption used for generation using the *pytorch-fid* repository. To align the settings of text-to-image evaluation with Rombach et al. (2022), we resize the short edge of the image to 256 and center crop a 256×256 patch before feeding them into the InceptionV3. We keep these settings for all samplers in our experiments.

## F.3 Baseline Settings

We mainly choose the three SOTA methods DPM-Solver (Lu et al., 2022a), DPM-Solver++(Lu et al., 2022b) and UniPC (Zhao et al., 2023) as our baselines. For the evaluation of DPM-Solver and DPM-Solver++, we use the code at `https://github.com/LuChengTHU/dpm-solver` and choose DPM-Solver-2S, DPM-Solver-3S, DPM-Solver++(2M) and DPM-Solver++(3M) as baselines. For UniPC, we use the code at `https://github.com/wl-zhao/UniPC` and choose UniPC-2-$B_1(h)$, UniPC-2-$B_2(h)$, UniPC-3-$B_1(h)$, UniPC-3-$B_2(h)$ and UniPC$_v$-3 as baselines. We follow Lu et al. (2022b) to choose uniform $t$ as the time schedule for high-resolution datasets (i.e., ImageNet-128, ImageNet-256 and LSUN-Bedroom) and text-image generation. We use lower orders at the final few steps for DPM-Solver++ and UniPC following their implementation. We keep other settings as default in the official implementations of these samplers.

## F.4 Search Algorithm Settings

We sample and evaluated 4000 solver schedules on CIFAR-10, 3000 solver schedules on CelebA and ImageNet-64, 2000 solver schedules ImageNet-128 and ImageNet-256, and 1500 solver schedules on LSUN-Bedroom and for text-to-image generation. We sample 1000 images to evaluate the true performance of all solver schedules. We use 3 stages in the multi-stage search process, with the first stage evaluating half of all schedules and the other two stages each evaluating a quarter of all schedules.

## F.5 Discussion of Search Overheads

Since the we need to generate images for solver schedule evaluation, calculate the metric and train predictors, our method introduces additional overhead. The network used to calculate FID is much lighter than diffusion U-Net and need not to inference iteratively. The predictor is even lighter and its training costs only tens of minutes. Therefore, the total cost is mainly on the generation of images. This overhead is equal to the time cost of N×M×S times neural inference of the diffusion U-Net (together with the classifier if it is classifier-guidance sampling), where N is the number of evaluated solver schedules, M is the number of images generated to calculate the metric, and S is the mean NFE of all solver schedules. N and M for each dataset can be found at App. F.4. Since we conduct the search under NFE budget 4-10 and we uniformly sample budget in this range, we use the mean value 7 for S. We then test the runtime for diffusion U-Net to forward one batch of images on one single GPU and estimate the overall GPU time cost. We list the results in Tab. 6.

Table 6: Computational Cost of $S^3$. N stands for the total number of evaluated solver schedules. M stands for the number of generated images to calculate the metric. S stands for the mean NFE of all solve schedules. Time/Batch is the GPU time consumed for the diffusion U-Net to complete one time of inference with the corresponding batch size. Total GPU hours is the overall overhead of the search, which is estimated by N×M×S×(Time/Batch)/3600. "Ours-250" stands for our ablated setting of using only 250 images to calculate the metric, whose performance can be found at Tab. 3.

| Dataset | N | M | S | Time/Batch | Total GPU hours | Device |
|---|---|---|---|---|---|---|
| CIFAR-10 | 4000 | 1000 | 7 | 400ms/256 | 12.15 | NVIDIA 3090 |
| CelebA | 3000 | 1000 | 7 | 500ms/256 | 11.39 | NVIDIA 3090 |
| ImageNet-64 | 3000 | 1000 | 7 | 788ms/256 | 17.96 | NVIDIA 3090 |
| ImageNet-128 | 2000 | 1000 | 7 | 2465ms/256 | 37.45 | NVIDIA A100 |
| ImageNet-256 | 2000 | 1000 | 7 | 2148ms/64 | 130.33 | NVIDIA A100 |
| LSUN-Bedroom | 1500 | 1000 | 7 | 3332ms/128 | 68.36 | NVIDIA A100 |
| MS-COCO | 1500 | 1000 | 7 | 585ms/16 | 106.64 | NVIDIA A100 |
| MS-COCO (Ours-250) | 1500 | 250 | 7 | 585ms/16 | 26.66 | NVIDIA A100 |

**Comparison with other methods** We compare our overhead with a popular training-based method Consistency Model (Song et al., 2023) and a concurrent work DPM-Solver-v3 (Zheng et al., 2023) of improving training-free sampler to demonstrate the efficiency of our search method $S^3$.

- **Training-based Method.** While distillation-based methods have astonishing performance with very low NFE, their overheads are far larger than ours. Consistency model (Song et al., 2023) is a popular and effective work which achieves very impressive performance with 1-4 sampling steps. However, the training cost of this work is heavy. According to Tab.3 in its paper, consistency distillation/training on CIFAR-10 needs 800k×512 samples, which is nearly 15 times more than 4000×1000×7 in our methods. Moreover, one iteration of training consistency models needs more than one forward pass (3 for CD and 2 for CT) and an additional backward process (nearly 3× time cost than forward pass). Therefore, our search process is nearly 90 times faster than training consistency models. For large resolution datasets, this ratio is even larger. On LSUN-Bedroom, consistency distillation needs 600k×2048 samples, which is 117 times more than 1500×1000×7 in our methods, and the overall cost is almost 700 times larger. Besides, our method does not have GPU memory limitations, while training based methods suffer from this problem (e.g., consistency models are trained on 8 GPUs on CIFAR10 and 64 GPUs on LSUN). In conclusion, our $S^3$ offers a effective trade-off between sample quality and computation overhead compared to training-based methods.

- **Training-free Sampler.** DPM-Solver-v3 (Zheng et al., 2023) is a concurrent work that aims to improve the training-free diffusion samplers. It replaces traditional model prediction types (i.e., data prediction and noise prediction) with a novel new prediction type. To calculate such a prediction type, DPM-Solver-v3 has to calculate empirical model statistics (EMS) related to the data distribution before being used for sampling. Therefore, similar to $S^3$, this method also needs additional offline overhead for the EMS calculation. According to Appendix D in this paper, we compare its cost with our $S^3$ search in Tab. 7. We can see that the overhead of these two methods is within the same order of magnitude. It is worth mentioning that $S^3$ can incorporate DPM-Solver-v3 by introducing its novel prediction type as another searchable strategy without changes to other components and be further boosted.

## G ADDITIONAL RESULTS

### G.1 COMPARISON WITH MORE BASELINE METHODS

We provide the full results of all baseline methods at Tabs. 8 to 14. We report best and worst performance of these baselines in Sec. 5.

Table 7: Comparison of the additional cost (GPU hours) between DPM-Solver-v3 (Zheng et al., 2023) and our method. We use the setting of 250 generated images for "Ours-250" and the results can be found at Tab. 3 and Tab. 16

| Dataset | CIFAR10 | MS-COCO |
|---------|---------|---------|
| DPM-Solver-v3 | 56 | 88 |
| Ours | 12.15 | 106.64 |
| Ours-250 | **3.04** | **26.66** |

Table 8: Full baseline results on CIFAR-10

| Method | NFE | | | | | | |
|--------|-----|-----|-----|-----|-----|-----|-----|
| | 4 | 5 | 6 | 7 | 8 | 9 | 10 |
| DPM-Solver-2S | 247.23 | 32.16 | 32.15 | 13.22 | 10.80 | 6.41 | 5.97 |
| DPM-Solver-3S | 255.21 | 288.12 | 23.87 | 14.79 | 22.99 | 5.72 | 4.70 |
| DPM-Solver++(2M) | 61.13 | 33.85 | 20.84 | 13.89 | 10.34 | 7.98 | 6.76 |
| DPM-Solver++(3M) | 61.13 | 29.39 | 13.18 | 7.18 | 5.27 | 4.41 | 3.99 |
| UniPC-2-$B_1(h)$ | 62.67 | 33.24 | 16.78 | 9.96 | 6.87 | 5.19 | 4.41 |
| UniPC-2-$B_2(h)$ | 60.20 | 31.10 | 17.77 | 11.23 | 8.03 | 6.27 | 5.40 |
| UniPC-3-$B_1(h)$ | 62.67 | 23.44 | 10.33 | 6.47 | 5.16 | 4.30 | 3.91 |
| UniPC-3-$B_2(h)$ | 60.20 | 26.30 | 11.64 | 6.83 | 5.18 | 4.31 | 3.90 |
| UniPC$_v$-3 | 57.52 | 24.93 | 11.12 | 6.67 | 5.15 | 4.30 | 3.90 |
| Ours | **11.50** | **6.86** | **5.18** | **3.81** | **3.41** | **3.02** | **2.69** |

## G.2 SEARCH RESULTS UNDER LARGER NFE BUDGETS

To further show the effectiveness of our method, we report the FID result under larger NFE budgets in Tab. 15. Noting that when the NFE budget is adequate, the negative impact of sub-optimal empirical strategies diminishes. So, it is nothing surprising that the gap between our method and baselines decreases. But we can see that our method still outperforms all baselines by a large margin at larger NFEs like 12, 15, and 20. Our method completely converges at NFE=15, much faster than existing solvers.

## G.3 MORE RESULTS WITH LOWER SEARCH COST

Since $S^3$ has additional search cost, it is meaningful to investigate its performance if the search cost is limited. As discussed in Sec. 5.2 and App. F.5, the search overhead of $S^3$ is proportional to the number of evaluated schedules and the number of images for FID calculation. In Sec. 5.2, we demonstrate the results of using less generated images to calculate the metric. In this section, we conduct more systematic experiments on CIFAR-10 to investigate the impact of both the number of sampled images and the number of evaluated solver schedules. Our results are listed in Tab. 16. From these results, we can see that though in most cases, lower search cost leads to worse search results, our method consistently outperforms the best baseline method significantly. These results show the large space of optimization for baseline solver schedules, requiring only small search cost to enhance.

## G.4 SEARCH RESULTS WITH OTHER SEARCH METHODS AND SPACES

In addition to the search space we introduce in App. B and the multi-stage method we introduce in Sec. 4.2, we try other search space designs and search approaches. We introduce these methods in this section.

### G.4.1 SEARCH METHOD

**Cascade Search** The search method introduced in Sec. 4.2 searches for all components simultaneously, which may lead to a huge search space and thus is inefficient. We try another cascade search

Table 9: Full baseline results on CelebA

| Method | NFE | | | | | | |
|---|---|---|---|---|---|---|---|
| | 4 | 5 | 6 | 7 | 8 | 9 | 10 |
| DPM-Solver-2S | 313.62 | 31.14 | 52.04 | 7.61 | 7.40 | 4.98 | 4.37 |
| DPM-Solver-3S | 321.39 | 330.10 | 25.14 | 17.28 | 16.99 | 10.39 | 6.91 |
| DPM-Solver++(2M) | 31.27 | 20.37 | 14.18 | 11.16 | 9.28 | 8.00 | 7.11 |
| DPM-Solver++(3M) | 31.27 | 14.31 | 8.21 | 7.56 | 6.43 | 5.17 | 4.47 |
| UniPC-2-$B_1(h)$ | 28.51 | 13.90 | 8.95 | 7.44 | 5.96 | 4.84 | 4.20 |
| UniPC-2-$B_2(h)$ | 29.95 | 18.57 | 12.29 | 9.83 | 7.74 | 6.81 | 5.97 |
| UniPC-3-$B_1(h)$ | 28.51 | 8.38 | 6.72 | 6.72 | 5.17 | 4.21 | 4.02 |
| UniPC-3-$B_2(h)$ | 29.95 | 12.44 | 7.84 | 7.38 | 5.92 | 4.71 | 4.25 |
| UniPC$_v$-3 | 26.32 | 10.61 | 7.31 | 7.13 | 5.68 | 4.55 | 4.15 |
| Ours | **12.31** | **5.17** | **3.65** | **3.80** | **3.62** | **3.16** | **2.73** |

Table 10: Full baseline results on ImageNet-64

| Method | NFE | | | | | | |
|---|---|---|---|---|---|---|---|
| | 4 | 5 | 6 | 7 | 8 | 9 | 10 |
| DPM-Solver-2S | 321.67 | 65.16 | 72.47 | 34.08 | 35.00 | 28.22 | 27.99 |
| DPM-Solver-3S | 364.60 | 366.66 | 51.27 | 47.84 | 54.21 | 23.89 | 24.76 |
| DPM-Solver++(2M) | 82.72 | 63.48 | 50.35 | 40.99 | 34.80 | 30.56 | 27.96 |
| DPM-Solver++(3M) | 82.72 | 69.06 | 45.32 | 33.25 | 27.65 | 25.47 | 24.56 |
| UniPC-2-$B_1(h)$ | 93.98 | 64.38 | 49.12 | 36.12 | 29.42 | 26.04 | 24.23 |
| UniPC-2-$B_2(h)$ | 80.70 | 61.73 | 47.61 | 38.03 | 31.96 | 28.08 | 25.79 |
| UniPC-3-$B_1(h)$ | 93.87 | 67.32 | 41.73 | 31.45 | 27.21 | 25.67 | 24.76 |
| UniPC-3-$B_2(h)$ | 80.73 | 69.08 | 43.80 | 32.05 | 26.98 | 25.26 | 24.35 |
| UniPC$_v$-3 | 76.69 | 67.05 | 42.81 | 31.76 | 26.99 | 25.34 | 24.44 |
| Ours | **33.84** | **24.95** | **22.31** | **19.55** | **19.19** | **19.09** | **16.68** |

method. We start by searching for only some of the components and keep other components consistent with the setting of baselines. After obtaining the search results with this sub-search space, we search for other components based on these searched schedules and keep the rest of the components unsearchable. In our experiment, we search for timesteps, Taylor expansion orders, prediction types, and correctors at the first stage, then search for the derivative estimation method in the second stage. The results of this search method are shown in Tab. 17. The number of sampled schedules is kept consistent for both search methods. Although the cascade method is more likely to fall into a local optimal, our results show that under some NFE budgets, the cascade method finds better solver schedules, indicating that it can also be used for searching.

### G.4.2 SEARCH SPACES

**Starting Point (SP)** Since singlestep methods usually perform worse than multistep methods, we don't include other choices of starting points than the point before the target timestep in the original search space used by experiments in Sec. 5. To explore the potential of a more distant starting point, we add the second point before the target timestep into the original search space and obtain the results shown in Tab. 18 (see the row 'Ours-SP'). We keep the search method and settings constant. We find that though this new search space provides more potential to find better solver schedules under some NFE budgets, the search results under most budgets are worse than the original search space. This is because the newly added decision does not bring improvements in most cases, making the search process more inefficient. Therefore, we recommend using multistep solver when the NFE budget is very tight.

**Interpolated Prediction Type (IP).** There are only two choices for prediction type in our original search space. We further try an interpolation between these two types given below:

$$x_t = ax_t^x + (1 - a)x_t^\epsilon$$

Table 11: Full baseline results on LSUN-Bedroom

| Method | NFE | | | | | | |
|---|---|---|---|---|---|---|---|
| | 4 | 5 | 6 | 7 | 8 | 9 | 10 |
| DPM-Solver++(2M) | 44.29 | 24.33 | 15.96 | 12.41 | 10.41 | 9.28 | 8.49 |
| DPM-Solver++(3M) | 44.29 | 20.79 | 13.27 | 10.85 | 9.62 | 8.95 | 8.38 |
| UniPC-2-$B_1(h)$ | 38.66 | 19.09 | 13.88 | 12.06 | 10.87 | 9.99 | 9.22 |
| UniPC-2-$B_2(h)$ | 22.02 | 20.60 | 13.79 | 11.27 | 9.92 | 9.11 | 8.52 |
| UniPC-3-$B_1(h)$ | 38.66 | 17.98 | 13.09 | 11.49 | 10.47 | 9.66 | 8.89 |
| UniPC-3-$B_2(h)$ | 22.02 | 18.44 | 12.44 | 10.69 | 9.76 | 9.14 | 8.53 |
| UniPC$_v$-3 | 39.14 | 17.99 | 12.43 | 10.79 | 9.97 | 9.26 | 8.60 |
| Ours | **16.45** | **12.98** | **8.97** | **6.90** | **5.55** | **3.86** | **3.76** |

Table 12: Full baseline results on ImageNet-128 (classifier-guidance generation, $s = 4.0$)

| Method | NFE | | | | | | |
|---|---|---|---|---|---|---|---|
| | 4 | 5 | 6 | 7 | 8 | 9 | 10 |
| DPM-Solver++(2M) | 26.59 | 14.42 | 10.08 | 8.37 | 7.50 | 7.06 | 6.80 |
| DPM-Solver++(3M) | 26.58 | 15.02 | 9.30 | 7.24 | 6.54 | 6.44 | 6.37 |
| UniPC-2-$B_1(h)$ | 32.08 | 15.39 | 9.01 | 7.13 | 6.62 | 6.49 | 6.36 |
| UniPC-2-$B_2(h)$ | 25.77 | 13.16 | 8.89 | 7.49 | 6.88 | 6.68 | 6.50 |
| UniPC-3-$B_1(h)$ | 32.08 | 16.14 | 9.81 | 7.20 | 6.28 | 6.03 | 6.03 |
| UniPC-3-$B_2(h)$ | 25.77 | 14.68 | 9.57 | 7.24 | 6.28 | 6.06 | 6.04 |
| UniPC$_v$-3 | 26.60 | 14.82 | 9.55 | 7.31 | 6.35 | 6.12 | 6.03 |
| Ours | **18.61** | **8.93** | **6.68** | **5.71** | **5.28** | **4.81** | **4.69** |

where $x_t^x$ and $x_t^\epsilon$ are given by:

$$x_t^\epsilon = \frac{\alpha_t}{\alpha_s} x_s - \sigma_t \sum_{k=0}^{n} h^{k+1} \varphi_{k+1}^\epsilon(h) \epsilon_\theta^{(k)}(x_s, s),$$

$$x_t^x = \frac{\sigma_t}{\sigma_s} x_s + \alpha_t \sum_{k=0}^{n} h^{k+1} \varphi_{k+1}^x(h) x_\theta^{(k)}(x_s, s)$$

Note that interpolating the neural term by $f_\theta(x_t, t) = ax_\theta(x_t, t) + (1-a)\epsilon_\theta(x_t, t)$ will lead to an integral term which can not be analytically computed as the integral in Eq. (36), causing a more complicate situation, so we simply interpolate the solution of these two prediction types. We search for the interpolation ratio $a$ only and evaluate another 500 schedules based on the search results of the cascade search method (see App. G.4.1) on CIFAR-10. The results are shown in the 'Ours-IP' row of Tab. 19. We can see that the continuous search space of prediction type may have the potential to boost the search results with low NFE budget like 4, 5, or 6.

**Guidance Scale (GS).** The choice of guidance scale for classifier-free guidance has a significant impact on the content and quality of the generated image. Usually, guidance scale is given by the users to adjust the control strength of provided conditions. But in some cases, users only care about the image quality and have to try different values of guidance scale to get high-quality samples. Therefore, we propose to include this hyper-parameter in our search space. Like other strategies, we allow the guidance scale of each timestep to be searched independently. Our search results are shown at Tab. 20. From the results of baseline methods with different guidance scales, we validate the fact that a suitable guidance scale is very important for sample quality, and our $S^3$ can help find proper guidance scales to achieve higher quality generation. Noting that sample quality evaluated by different metrics may have different preferences on guidance scale, but using $S^3$ can help to automatically find the optimal setting given arbitrary metrics.

Table 13: Full baseline results on ImageNet-256 (classifier-guidance generation, $s = 8.0$)

| Method | NFE | | | | | | |
|---|---|---|---|---|---|---|---|
| | 4 | 5 | 6 | 7 | 8 | 9 | 10 |
| DPM-Solver++(2M) | 51.09 | 27.71 | 17.62 | 13.19 | 10.91 | 9.85 | 9.31 |
| DPM-Solver++(3M) | 51.09 | 38.08 | 26.30 | 18.33 | 13.52 | 11.35 | 9.97 |
| UniPC-2-$B_1(h)$ | 80.46 | 49.10 | 29.91 | 19.70 | 14.38 | 11.69 | 10.20 |
| UniPC-2-$B_2(h)$ | 56.06 | 31.79 | 20.16 | 14.66 | 11.71 | 10.28 | 9.51 |
| UniPC-3-$B_1(h)$ | 80.46 | 54.00 | 38.67 | 29.35 | 22.06 | 16.74 | 13.66 |
| UniPC-3-$B_2(h)$ | 56.06 | 43.13 | 34.37 | 27.45 | 20.42 | 16.00 | 13.01 |
| Ours | **33.84** | **19.06** | **13.00** | **10.31** | **9.72** | **9.06** | **9.06** |

Table 14: Full baseline results of text-to-image generation on MS-COCO2014 validation set, $s = 8.0$)

| Method | NFE | | | | | | |
|---|---|---|---|---|---|---|---|
| | 4 | 5 | 6 | 7 | 8 | 9 | 10 |
| DPM-Solver-2S | 149.55 | 88.53 | 79.46 | 41.50 | 37.07 | 23.22 | 20.74 |
| DPM-Solver-3S | 161.03 | 156.72 | 106.15 | 75.28 | 58.54 | 39.26 | 29.54 |
| DPM-Solver++(2M) | 24.95 | 20.59 | 19.07 | 18.17 | 17.84 | 17.53 | 17.31 |
| DPM-Solver++(3M) | 24.95 | 20.59 | 18.80 | 17.89 | 17.54 | 17.42 | 17.22 |
| UniPC-2-$B_1(h)$ | 30.77 | 22.71 | 19.66 | 18.45 | 18.00 | 17.65 | 17.54 |
| UniPC-2-$B_2(h)$ | 25.49 | 20.65 | 19.16 | 18.13 | 17.84 | 17.65 | 17.52 |
| UniPC$_v$-3 | 26.39 | 21.30 | 19.05 | 17.83 | 17.58 | 17.50 | 17.53 |
| Ours | **22.76** | **16.84** | **15.76** | **14.77** | **14.23** | **13.99** | **14.01** |

## G.5 Transfer Ability Validation of Solver Schedules Searched on Text-to-image Task

To verify practicality of $S^3$, we use a different version model, SD-v1-3, to evaluate the solver schedules searched with SD-v1-5 model. Other settings are kept consistent with previous experiments in Sec. 5. We list the results in Tab. 21. We can see that the searched results can directly transfer to SD-v1-3 and achieve similar acceleration ratio with SD-v1-5 model, making it more convenient to apply our searched results.

## G.6 Searched Solver Schedules

We give some of our searched solver schedules on CIFAR-10 here as examples. We list these schedules in Tab. 22 and Tab. 23

## H Additional Derivations

In this section, we write the detailed derivation of Eq. (8) and Eq. (9) from Eq. (4) and Eq. (5), correspondingly.

### H.1 Noise Prediction Model

Firstly, we use the *variation of constant* method to derive Eq. (6). Suppose the solution $x(t)$ of Eq. (4) can be written as the product of two independent variables: $x(t) = u(t)v(t)$. Substitute it to Eq. (4), we get:

$$u(t)\mathrm{d}v + v(t)\mathrm{d}u = (f(t)u(t)v(t) + \frac{g^2(t)}{2\sigma_t}\epsilon_\theta(x_t, t))\mathrm{d}t. \tag{25}$$

Table 15: FIDs of searched solver schedules and baseline methods under larger NFE budgets.

| Dataset | Method | NFE | | |
|---|---|---|---|---|
| | | 12 | 15 | 20 |
| **CIFAR-10** | Baseline-W | 5.31 | 4.52 | 3.54 |
| | Baseline-B | 3.67 | 3.03 | 2.80 |
| | Ours | **2.65** | **2.41** | **2.41** |
| **CelebA** | Baseline-W | 6.12 | 4.20 | 3.56 |
| | Baseline-B | 3.82 | 2.72 | 2.70 |
| | Ours | **2.32** | **2.06** | **2.06** |

Table 16: FIDs of the searched solver schedules with lower search cost.

| Number of Schedule | Number of Image | NFE | | | | | | |
|---|---|---|---|---|---|---|---|---|
| | | 4 | 5 | 6 | 7 | 8 | 9 | 10 |
| **2000** | 1000 | 14.94 | **6.86** | **5.10** | **4.18** | 4.14 | **3.13** | **2.67** |
| | 500 | **11.50** | 6.94 | 5.11 | 4.40 | 3.95 | 3.20 | 2.90 |
| | 250 | 12.55 | 7.84 | 6.26 | 4.77 | **3.46** | **3.13** | 3.13 |
| **1500** | 1000 | 15.41 | 7.32 | 5.58 | 4.21 | 4.14 | 3.17 | 3.12 |
| | 500 | 11.97 | 7.33 | 5.11 | 4.40 | 4.12 | 3.20 | 3.20 |
| | 250 | 13.74 | 8.04 | 6.11 | 4.85 | **3.46** | 3.46 | 3.33 |
| **1000** | 1000 | 15.43 | 7.78 | 5.78 | 4.78 | 4.12 | 3.48 | 3.48 |
| | 500 | 11.97 | 8.01 | 5.31 | 4.40 | 4.41 | 3.20 | 3.20 |
| | 250 | 15.42 | 8.11 | 6.11 | 4.84 | **3.46** | 3.46 | 3.33 |
| Baseline-B | | 57.52 | 23.44 | 10.33 | 6.47 | 5.16 | 4.30 | 3.90 |

We let $u(t)\mathrm{d}v = f(t)u(t)v(t)$ to solve $v(t)$ and obtain $v(t) = v(s)e^{\int_s^t f(\tau)\mathrm{d}\tau}$, where $s$ is an arbitrary value larger than $t$. Then we substitute the $v(t)$ into Eq. (25):

$$v_t\mathrm{d}u_t = \frac{g^2(t)}{2\sigma_t}\epsilon_\theta(x_t, t)\mathrm{d}t. \tag{26}$$

Then we solve $u_t$ through Eq. (26).

$$u(t) = u(s) + \int_s^t \frac{g^2(\tau)}{2\sigma_\tau}\frac{\epsilon_\theta(x_\tau, \tau)}{v(\tau)}\mathrm{d}\tau \tag{27}$$

$$= u(s) + \int_s^t \frac{g^2(\tau)}{2\sigma_\tau}\frac{\epsilon_\theta(x_\tau, \tau)}{v(s)e^{\int_s^\tau f(r)\mathrm{d}r}}\mathrm{d}\tau \tag{28}$$

$$\tag{29}$$

Finally, we write $x(t) = u(t)v(t)$:

$$x(t) = u(t)v(t) = v(s)e^{\int_s^t f(r)\mathrm{d}r}\left(u(s) + \int_s^t \frac{g^2(\tau)}{2\sigma_\tau}\frac{\epsilon_\theta(x_\tau, \tau)}{v(s)e^{\int_s^\tau f(r)\mathrm{d}r}}\mathrm{d}\tau\right) \tag{30}$$

$$= v(s)e^{\int_s^t f(r)\mathrm{d}r}u(s) + \int_s^t \frac{g^2(\tau)}{2\sigma_\tau}\epsilon_\theta(x_\tau, \tau)e^{\int_\tau^t f(r)\mathrm{d}r}\mathrm{d}\tau \tag{31}$$

$$= x(s)e^{\int_s^t \mathrm{dlog}\alpha_r} + \int_s^t \frac{\frac{\mathrm{d}\sigma_\tau^2}{\mathrm{d}\tau} - 2\frac{\mathrm{dlog}\alpha_\tau}{\mathrm{d}\tau}\sigma_\tau^2}{2\sigma_\tau}\epsilon_\theta(x_\tau, \tau)e^{\int_\tau^t \mathrm{dlog}\alpha_r}\mathrm{d}\tau \tag{32}$$

$$= \frac{\alpha(t)}{\alpha(s)}x(s) - \alpha(t)\int_s^t \frac{\sigma_\tau}{\alpha_\tau}\epsilon_\theta(x_\tau, \tau)\mathrm{dlog}\frac{\alpha_\tau}{\sigma_\tau} \tag{33}$$

$$= \frac{\alpha(t)}{\alpha(s)}x(s) - \alpha(t)\int_{\lambda_s}^{\lambda_t} e^{-\lambda}\epsilon_\theta(x_\lambda, \lambda)\mathrm{d}\lambda. \tag{34}$$

Table 17: FIDs of the searched solver schedules using the cascade search method.

| Dataset | Method | NFE | | | | | | |
|---|---|---|---|---|---|---|---|---|
| | | 4 | 5 | 6 | 7 | 8 | 9 | 10 |
| CIFAR-10 | Baseline-W(S) | 255.21 | 288.12 | 32.15 | 14.79 | 22.99 | 6.41 | 5.97 |
| | Baseline-W(M) | 61.13 | 33.85 | 20.84 | 13.89 | 10.34 | 7.98 | 6.76 |
| | Baseline-B | 57.52 | 23.44 | 10.33 | 6.47 | 5.16 | 4.30 | 3.90 |
| | Ours | **11.50** | **6.86** | **5.18** | 3.81 | **3.41** | **3.02** | **2.69** |
| | Ours-Cascade | 12.60 | 7.17 | 5.81 | **3.76** | 3.76 | 3.23 | 3.23 |

Table 18: FIDs of the searched solver schedules in the search space containing more choice starting point.

| Dataset | Method | NFE | | | | | | |
|---|---|---|---|---|---|---|---|---|
| | | 4 | 5 | 6 | 7 | 8 | 9 | 10 |
| CelebA | Baseline-W(S) | 321.39 | 330.10 | 52.04 | 17.28 | 16.99 | 10.39 | 6.91 |
| | Baseline-W(M) | 31.27 | 20.37 | 14.18 | 11.16 | 9.28 | 8.00 | 7.11 |
| | Baseline-B | 26.32 | 8.38 | 6.72 | 6.72 | 5.17 | 4.21 | 4.02 |
| | Ours | 12.31 | **5.17** | **3.65** | **3.80** | **3.62** | **3.16** | 2.73 |
| | Ours-SP | **10.76** | 7.30 | 4.26 | 4.09 | 3.81 | 3.48 | **2.39** |
| ImageNet-64 | Baseline-W(S) | 364.60 | 366.66 | 72.47 | 47.84 | 54.21 | 28.22 | 27.99 |
| | Baseline-W(M) | 93.98 | 69.08 | 50.35 | 40.99 | 34.80 | 30.56 | 27.96 |
| | Baseline-B | 76.69 | 61.73 | 42.81 | 31.76 | 26.99 | 23.89 | 24.23 |
| | Ours | **33.84** | **24.95** | **22.31** | **19.55** | 19.19 | 19.09 | **16.68** |
| | Ours-SP | 34.31 | 28.03 | 23.31 | 19.92 | **18.34** | **18.34** | 17.75 |

Then we get the exponential integral in Eq. (6). According to Asm. A.2, we further write the $\epsilon_\theta(x_\lambda, \lambda)$ with Taylor expansion $\epsilon_\theta(x_\lambda, \lambda) = \sum_k^n \epsilon_\theta^{(k)}(x_{\lambda_s}, \lambda_s) \frac{(\lambda - \lambda_s)^k}{k!}$ ($n < \mathcal{M}$) and substitute into Eq. (6):

$$x_t = \frac{\alpha_t}{\alpha_s} x_s - \alpha_t \sum_{k=0}^{n} \epsilon_\theta^{(n)}(x_{\lambda_s}, \lambda_s) \int_{\lambda_s}^{\lambda_t} e^{-\lambda} \frac{(\lambda - \lambda_s)^k}{k!} d\lambda + \mathcal{O}((\lambda_t - \lambda_s)^{n+2}) \tag{35}$$

$$= \frac{\alpha_t}{\alpha_s} x_s - \sigma_t \sum_{k=0}^{n} \epsilon_\theta^{(n)}(x_{\lambda_s}, \lambda_s) \int_{\lambda_s}^{\lambda_t} e^{-(\lambda - \lambda_t)} \frac{(\lambda - \lambda_s)^k}{k!} d\lambda + \mathcal{O}((\lambda_t - \lambda_s)^{n+2}). \tag{36}$$

Let $h = \lambda_t - \lambda_s$, and suppose $\varphi_{k+1}^\epsilon = \frac{1}{h^{k+1}} \int_{\lambda_s}^{\lambda_t} e^{-(\lambda - \lambda_t)} \frac{(\lambda - \lambda_s)^k}{k!} d\lambda$ ($k \geq 0$). Use partial integral to calculate $\varphi_{k+1}^\epsilon$:

$$\varphi_{k+1}^\epsilon = \frac{1}{h^{k+1}} \left( -\frac{h^k}{k!} + \int_{\lambda_s}^{\lambda_t} e^{-(\lambda - \lambda_t)} \frac{(\lambda - \lambda_s)^{(k-1)}}{(k-1)!} d\lambda \right) \tag{37}$$

$$= \frac{\varphi_k^\epsilon(h) - 1/k!}{h}. \tag{38}$$

The initial value $\varphi_1^\epsilon(h)$ can be easily computed: $\varphi_1^\epsilon(h) = \frac{1}{h} \int_{\lambda_s}^{\lambda_t} e^{-(\lambda - \lambda_t)} d\lambda = \frac{e^h - 1}{h}$. And we further give $\varphi_0^\epsilon(h) = e^h$ through $\varphi_1^\epsilon = \frac{\varphi_0^\epsilon(h) - 1/0!}{h}$. By substituting $\varphi_{k+1}^\epsilon$ to Eq. (36), we obtain Eq. (8).

## H.2 DATA PREDICTION MODEL

The derivation with data prediction model is very similar to that with noise prediction model. Therefore, we omit some derivation in this section.

Table 19: FIDs of the searched solver schedules in the search space containing continuous interpolation ratio of prediction types.

| Dataset | Method | NFE | | | | | | |
|---------|--------|-----|-----|-----|-----|-----|-----|-----|
| | | 4 | 5 | 6 | 7 | 8 | 9 | 10 |
| | Baseline-W(S) | 255.21 | 288.12 | 32.15 | 14.79 | 22.99 | 6.41 | 5.97 |
| | Baseline-W(M) | 61.13 | 33.85 | 20.84 | 13.89 | 10.34 | 7.98 | 6.76 |
| CIFAR-10 | Baseline-B | 57.52 | 23.44 | 10.33 | 6.47 | 5.16 | 4.30 | 3.90 |
| | Ours | **11.50** | **6.86** | 5.18 | 3.81 | **3.41** | **3.02** | **2.69** |
| | Ours-Cascade | 12.60 | 7.17 | 5.81 | **3.76** | 3.76 | 3.23 | 3.23 |
| | Ours-IP | 12.47 | 6.92 | **4.90** | 3.77 | 3.77 | 3.27 | 3.27 |

Table 20: FIDs of the searched solver schedules in the search space containing guidance scale.

| Guidance Scale | Method | NFE | | | | | | |
|----------------|--------|-----|-----|-----|-----|-----|-----|-----|
| | | 4 | 5 | 6 | 7 | 8 | 9 | 10 |
| | Baseline-W(S) | 161.03 | 156.72 | 106.15 | 75.28 | 58.54 | 39.26 | 29.54 |
| 7.5 | Baseline-W(M) | 30.77 | 22.71 | 19.66 | 18.45 | 18.00 | 17.65 | 17.54 |
| | Baseline-B | 24.95 | 20.59 | 18.80 | 17.83 | 17.54 | 17.42 | 17.22 |
| | Ours | **22.76** | **16.84** | **15.76** | **14.77** | **14.23** | **13.99** | **14.01** |
| 3.0 | DPM-Solver++(3M) | 24.64 | 17.59 | 15.21 | 14.24 | 13.77 | 13.37 | 13.13 |
| | UniPC$_v$-3 | 22.01 | 16.60 | 14.88 | 14.14 | 13.73 | 13.42 | 13.20 |
| Searchable | Ours-GS | **15.63** | **13.94** | **13.90** | **13.69** | **12.81** | **12.56** | **12.34** |

Firstly, by using the *variation of constant* method in App. H.1, we can get the exponential integral with data prediction model:

$$x_t = \frac{\sigma_t}{\sigma_s} x_s + \sigma_t \int_{\lambda_s}^{\lambda_t} e^\lambda x_\theta(x_\lambda, \lambda) d\lambda. \tag{39}$$

Then we replace the $x_\theta(x_\lambda, \lambda)$ with its Taylor expansion:

$$x_t = \frac{\sigma_t}{\sigma_s} x_s + \sigma_t \sum_{k=0}^{n} x_\theta^{(k)}(x_{\lambda_s}, \lambda_s) \int_{\lambda_s}^{\lambda_t} e^\lambda \frac{(\lambda - \lambda_s)^k}{k!} d\lambda \tag{40}$$

$$= \frac{\sigma_t}{\sigma_s} x_s + \alpha_t \sum_{k=0}^{n} x_\theta^{(k)}(x_{\lambda_s}, \lambda_s) \int_{\lambda_s}^{\lambda_t} e^{\lambda - \lambda_t} \frac{(\lambda - \lambda_s)^k}{k!} d\lambda. \tag{41}$$

Let $h = \lambda_t - \lambda_s$, and suppose $\varphi_{k+1}^x = \frac{1}{h^{k+1}} \int_{\lambda_s}^{\lambda_t} e^{\lambda - \lambda_t} \frac{(\lambda - \lambda_s)^k}{k!} d\lambda$ ($k \geq 0$). Use partial integral to calculate $\varphi_{k+1}^x$:

$$\varphi_{k+1}^x = \frac{1}{h^{k+1}} \Big( \frac{h^k}{k!} - \int_{\lambda_s}^{\lambda_t} e^{\lambda - \lambda_t} \frac{(\lambda - \lambda_s)^{(k-1)}}{(k-1)!} d\lambda \tag{42}$$

$$= \frac{1/k! - \varphi_k^x(h)}{h}. \tag{43}$$

The initial value $\varphi_1^x(h)$ can be easily computed: $\varphi_1^x(h) = \frac{1}{h} \int_{\lambda_s}^{\lambda_t} e^{\lambda - \lambda_t} d\lambda = \frac{1 - e^{-h}}{h}$. And we further give $\varphi_0^x(h) = e^{-h}$ through $\varphi_1^x = \frac{1/0! - \varphi_0^x(h)}{h}$. By substituting $\varphi_{k+1}^x$ to Eq. (41), we obtain Eq. (9).

## I  DISCUSSION OF CONVERGENCE

In this section, we discuss the convergence order of USF. We give the convergence bound at Thm. I.1.

Table 21: Results on MS-COCO 256×256 with SD-V1-3 model.

| Method | NFE | | | | | | |
|---|---|---|---|---|---|---|---|
| | 4 | 5 | 6 | 7 | 8 | 9 | 10 |
| UniPC$_v$-3 | 24.50 | 21.09 | 19.04 | 18.06 | 17.71 | 17.64 | 17.64 |
| DPM-Solver++(3M) | 24.50 | 20.58 | 18.83 | 18.25 | 17.75 | 17.56 | 17.43 |
| Ours | **22.49** | **16.74** | **15.81** | **14.84** | **14.42** | **14.29** | **14.40** |

Table 22: Searched schedule under 4 and 7 NFE budget on CIFAR-10. "D" and "N" in the row of "Prediction Type" stand for "data prediction" and "noise prediction". "T" and "F" in the row of "Corrector" stand for using and not using corrector. The same for "T" and "F" in the row of "Low Order Estimation".

| NFE | 4 | 7 |
|---|---|---|
| Timestep | 1.0,0.57,0.27,0.10,5.5e-4 | 1.0,0.79,0.61,0.42,0.25,0.08,0.03,1.3e-3 |
| Order | 1,1,3,2 | 1,2,3,2,3,2,2 |
| Prediction Type | D,D,N,D | D,D,D,N,N,D,D |
| Corrector | T,T,F,F | T,T,T,T,T,F,F |
| Scaling | -,-,$R_1$,$R_3$ | -,$R_2$,$R_0$,$R_0$,$R_0$,$R_1$,$R_1$ |
| Low Order Estimation | -,-,F,- | -,-,F,-,F,-,- |

**Theorem I.1.** *Suppose $x_{t_i}$ is the ground truth trajectory value, and the estimated value $\tilde{x}_{t_i}(h)$ is given by the one step update in Alg. 1, where $h = \lambda_{t_i} - \lambda_{t_{s_i}}$ is the step length w.r.t. the logSNR. Then the convergence order of $\tilde{x}_{t_i}(h)$ to h is $p_i = min(n_i + 1, l_{s_i}, d_1 + 2, d_2 + 3, \cdots, d_{n_i-1} + n_i)$, where $l_{s_i}$ is the convergence order of the starting point and $d_k$ is the convergence order of the k-th derivative estimation $\tilde{\epsilon}_\theta^{(k)}(x_{t_{s_i}}, t_{s_i})$ or $\tilde{x}_\theta^{(k)}(x_{t_{s_i}}, t_{s_i})$ to h.*

The proof is very simple by substituting the starting point $\tilde{x}_{t_{s_i}} = x_{t_{s_i}} + \mathcal{O}(h^{l_{s_i}})$ and the estimation of all derivatives $\tilde{\epsilon}_\theta^{(k)}(x_{t_{s_i}}, t_{s_i}) = \epsilon_\theta^{(k)}(x_{t_{s_i}}, t_{s_i}) + \mathcal{O}(h^{d_k})$ into Eq. (8) (the same for data prediction model). As a result, the convergence order of USF is decided by the Taylor expansion order $n_i$, accuracy of the starting point and the convergence orders of all derivative estimations.

As discussed in App. B, when no low-order derivative estimation is used and the additional involved points have convergence orders not smaller than $n_i$, the convergence order of the k-th derivative estimation is $n_i - k$, and the scaling operation doesn't change the convergence order of the derivative estimation. Therefore, the convergence order $p_i$ of $\tilde{x}_{t_i}(h)$ equals $n_i + 1$. When low-order estimation is used, the convergence order $p_i$ will decreases. It is worth noting that the there is no strong relation with the convergence order and the actual truncation error. As shown in Fig. 2, using low-order derivative estimation in certain timesteps can achieve lower error with the ground truth value. This is because the actual truncation error is not smooth and using high-order information for low-order derivative may introduce additional bias. Therefore, we apply low-order Taylor-difference method for derivative estimation in our experiments.

## J  QUALITATIVE RESULTS

We provide some samples generated by our searched solver schedules and baseline methods in this section.

Table 23: Searched schedule under 9 NFE budget on CIFAR-10. "D" and "N" in the row of "Prediction Type" stand for "data prediction" and "noise prediction". "T" and "F" in the row of "Corrector" stand for using and not using corrector. The same for "T" and "F" in the row of "Low Order Estimation".

| NFE | 9 |
|---|---|
| Timestep | 1.0,0.76,0.57,0.43,0.32,0.30,0.16,0.057,0.023,6.3e-4 |
| Order | 1,2,3,2,3,3,3,1,3 |
| Prediction Type | D,N,D,D,D,D,D,D,D |
| Corrector | T,T,T,T,T,T,T,F,F |
| Scaling | -,$R_3$,$R_1$,$R_0$,-,$R_3$,$R_2$,$R_4$,$R_0$ |
| Low Order Estimation | -,-,T,-,T,T,T,-,F |

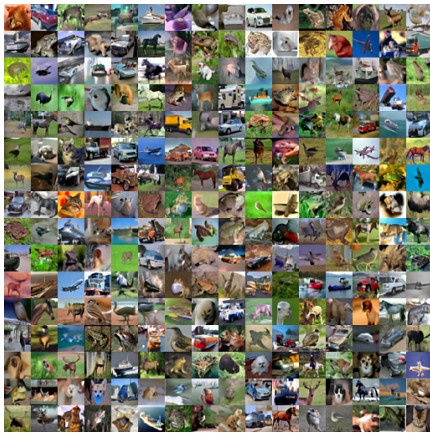
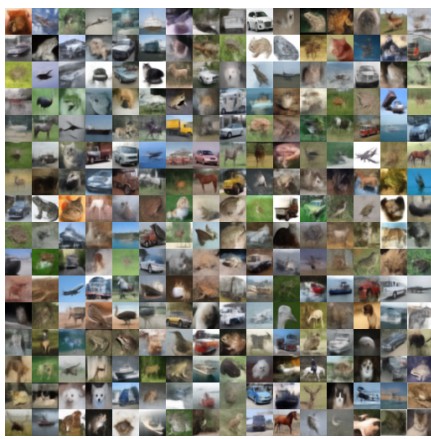

(a) Samples generated by the searched schedule    (b) Samples generated by baseline-B method

Figure 5: Samples of CIFAR-10 dataset with NFE=4.

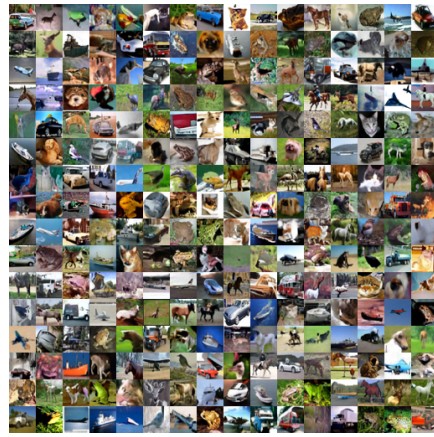
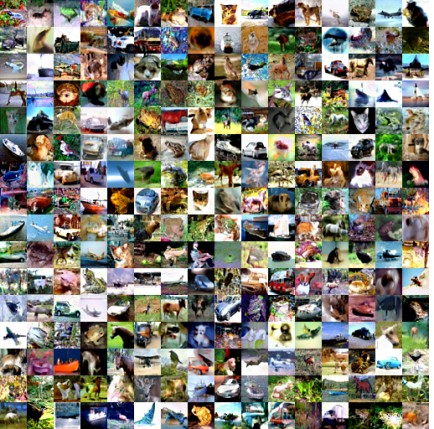

(a) Samples generated by the searched schedule    (b) Samples generated by baseline-B method

Figure 6: Samples of CIFAR-10 dataset with NFE=5.

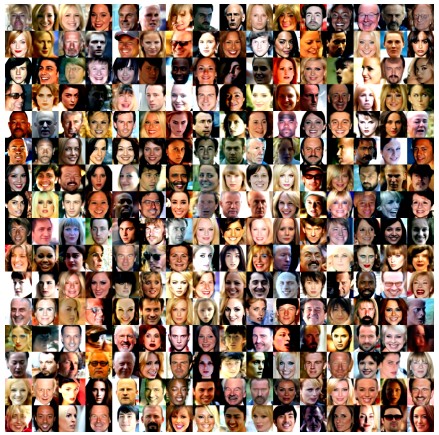 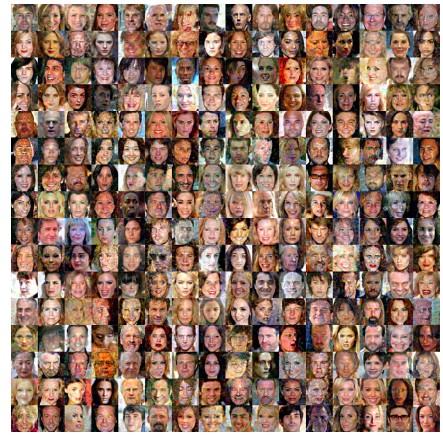

(a) Samples generated by the searched schedule        (b) Samples generated by baseline-B method

Figure 7: Samples of CelebA dataset with NFE=4.

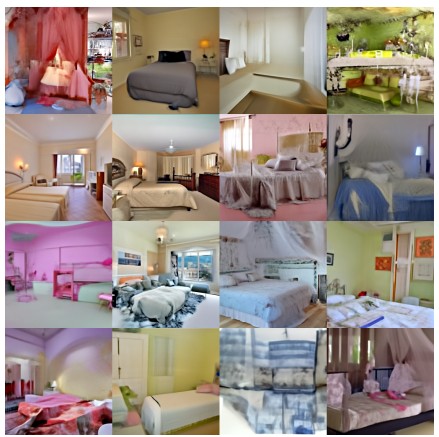 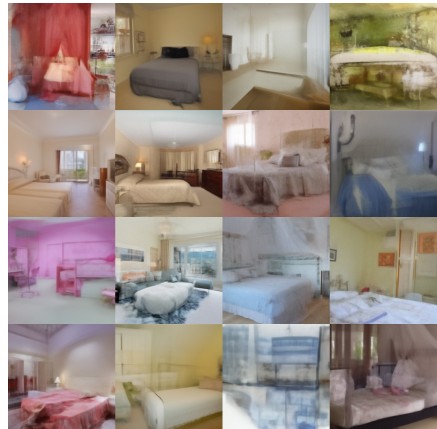

(a) Samples generated by the searched schedule        (b) Samples generated by baseline-B method

Figure 8: Samples of LSUN-Bedroom dataset with NFE=4.

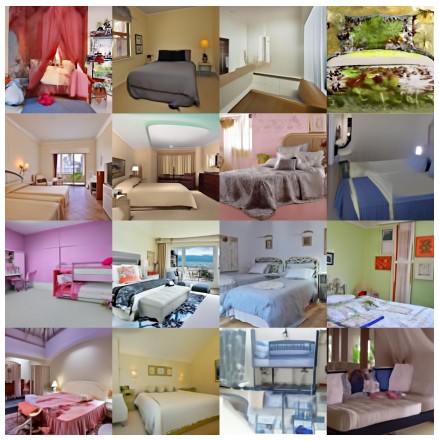 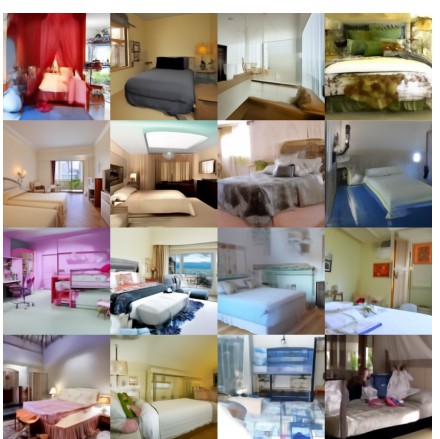

(a) Samples generated by the searched schedule        (b) Samples generated by baseline-B method

Figure 9: Samples of LSUN-Bedroom dataset with NFE=5.

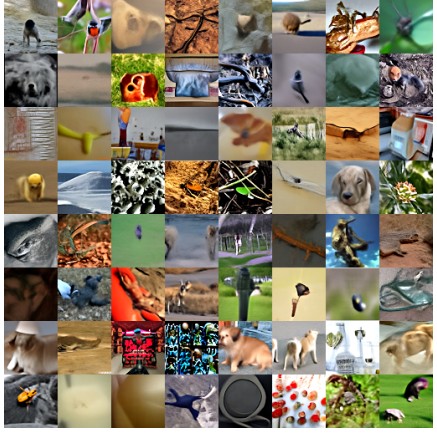
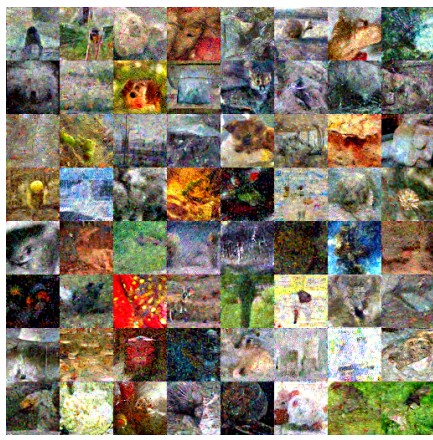

(a) Samples generated by the searched schedule      (b) Samples generated by baseline-B method

Figure 10: Samples of ImageNet-64 dataset with NFE=4.

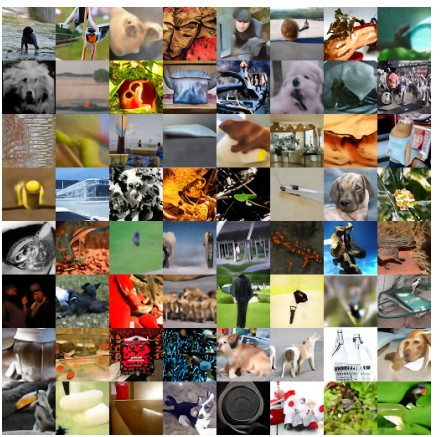
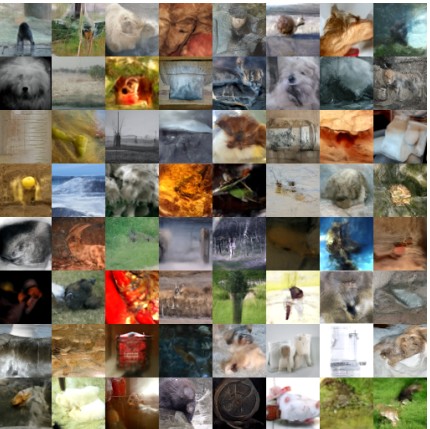

(a) Samples generated by the searched schedule      (b) Samples generated by baseline-B method

Figure 11: Samples of ImageNet-64 dataset with NFE=5.

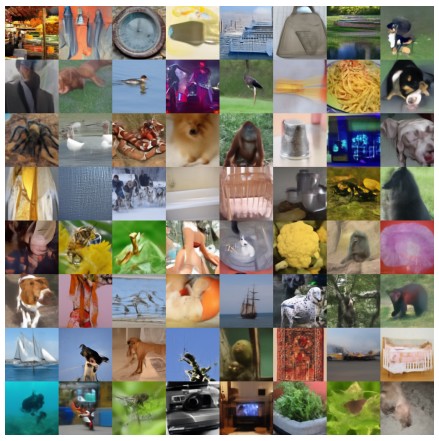
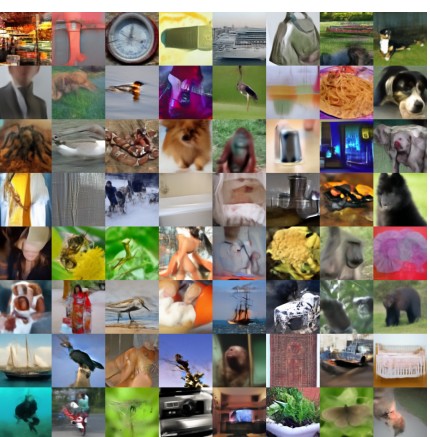

(a) Samples generated by the searched schedule      (b) Samples generated by baseline-B method

Figure 12: Samples of ImageNet-128 dataset with NFE=4.

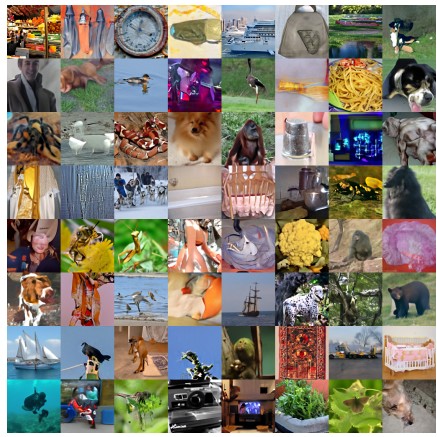 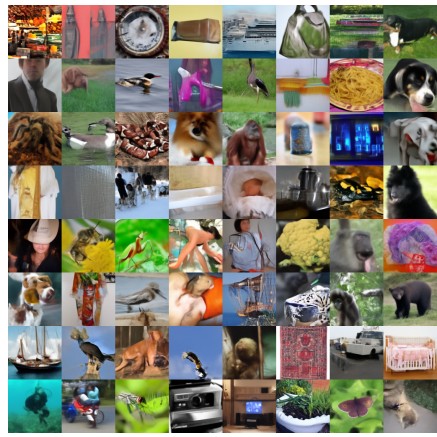

(a) Samples generated by the searched schedule        (b) Samples generated by baseline-B method

Figure 13: Samples of ImageNet-128 dataset with NFE=5.

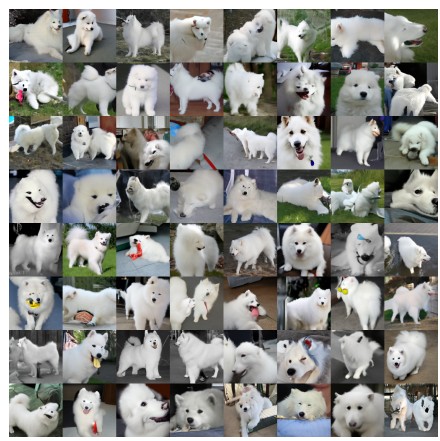 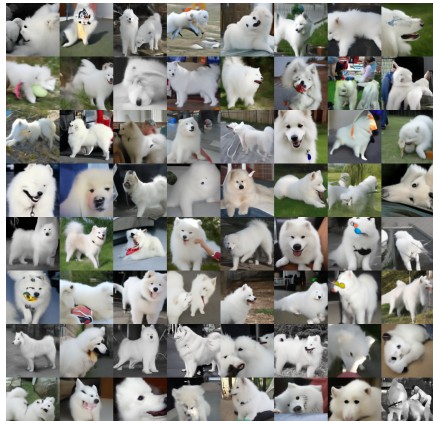

(a) Samples generated by the searched schedule        (b) Samples generated by baseline-B method

Figure 14: Samples of ImageNet-128 dataset with NFE=7.

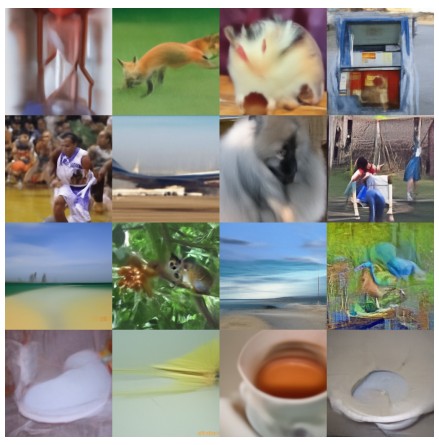 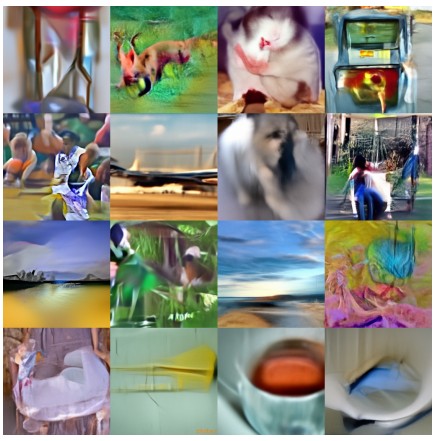

(a) Samples generated by the searched schedule        (b) Samples generated by baseline-B method

Figure 15: Samples of ImageNet-256 dataset with NFE=4.

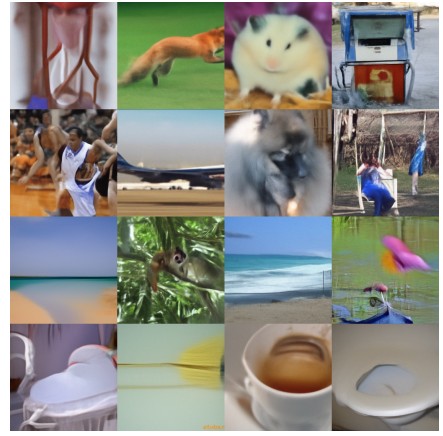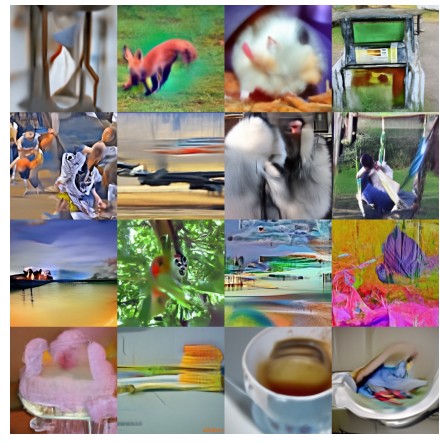

(a) Samples generated by the searched schedule      (b) Samples generated by baseline-B method

Figure 16: Samples of ImageNet-256 dataset with NFE=5.

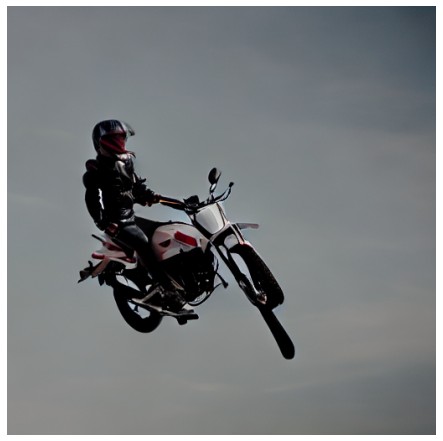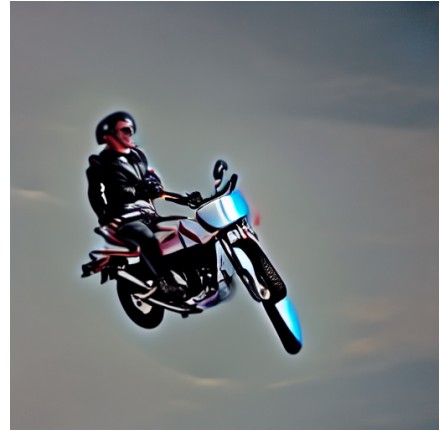

(a) Samples generated by the searched schedule      (b) Samples generated by baseline-B method

Figure 17: Samples guided by the prompt 'A person on a motorcycle high in the air' with NFE=5.

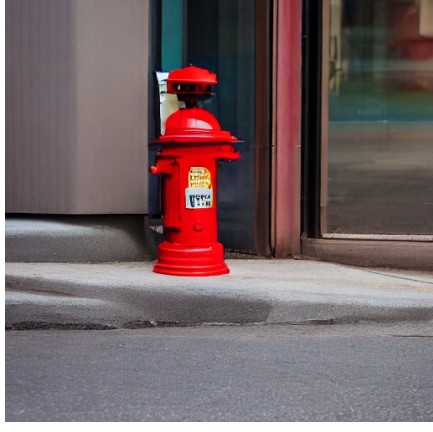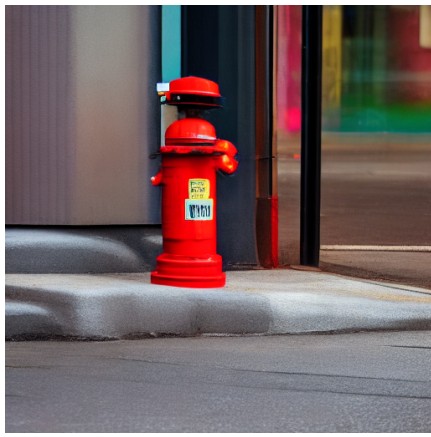

(a) Samples generated by the searched schedule      (b) Samples generated by baseline-B method

Figure 18: Samples guided by the prompt 'A fire hydrant sitting on the side of a city street' with NFE=5.

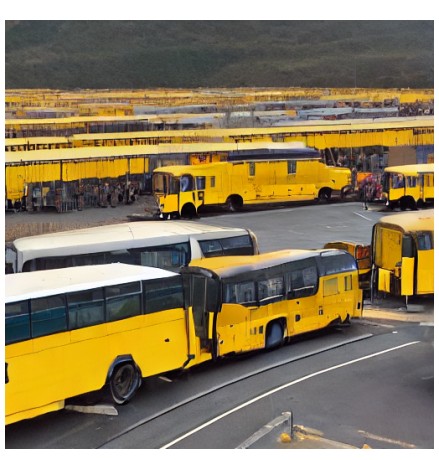
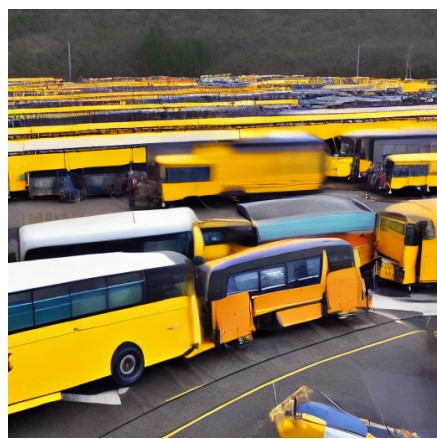

(a) Samples generated by the searched schedule   (b) Samples generated by baseline-B method

Figure 19: Samples guided by the prompt 'A bus yard filled with yellow school buses parked side by side' with NFE=5.

