# OpenReview forum: "A Unified Sampling Framework for Solver Searching of Diffusion Probabilistic Models"
_ICLR.cc/2024/Conference — ICLR 2024 poster_

### Official Review · Reviewer_qhfh · 2023-10-28

**Soundness:** 3 good
**Presentation:** 4 excellent
**Contribution:** 2 fair
**Rating:** 6
**Confidence:** 4

**Summary:**

The paper presents an AutoML-style experimental setup for searching optimal Diffusion Model sampler (solver). The authors, first noted the various “moving parts” of modern reverse process solver and then applied an evolution-based search technique to explore these while keeping track of FID under an NFE budget. Authors argued and validated that a non-uniform combination of these design choices in different timesteps is the key to reach optimal “solver schedule”. A lot of experiments are conducted to search through this space of solver schedules. In fact, they also proposes a special evolutionary search method S3 that does it more efficiently.

**Strengths:**

As far as I can tell, there hasn’t been such an effort to use AutoML setup for searching for better solver schedule — so it is indeed novel in some sense. Also, the large amount of experiments have been conducted, which can also be quite helpful for the community in general. The presentation/writing quality is also pretty good.

**Weaknesses:**

Here are some concerns I’d like to emphasize:

- The major weakness I see is that the paper can be called “just a search” without any solid conclusion. While I agree that figuring out good FID numbers for existing benchmarks is pretty good, it is not usually preffered to lock the contribution to very specific models/dataset/solver. If a new dataset/model/solver comes out in future, one need to perform the entire search again.
- The USF isn’t really something new. It is just a “dumb” aggregation of a bunch of design options. It is only based on empirical evidence — but I wonder if it can be theoretically justified.

**Questions:**

Here are some concrete questions I’d like the authors to answer.

- In high-level, it is not clear to me which part of the paper is being claimed to be the primary contribution — the search algorithm S3 itself, or the optimal solver schedules and corresponding FID numbers ? Is the paper supposed to be about the “search algorithm S3” or an empirical AutoML report ?
- If your focus is S3, there is surprisingly little written about it in the main paper. Almost all details are in supplementary. So that indicates it to be just an AutoML experimental report — is it ?
- Outright confession: I am not at all experienced in AutoML. So, is this S3 something novel or it is a usual practice in AutoML ? I don’t see any proper citation in case this has been adopted from existing literature/libraries.
- What exactly the paper “concludes” at the end the day — did you find any “pattern” in the optimal solver schedules that may motivate further work? I don’t see the exact configuration of the optimal solver schedules (found by S3) — are they written anywhere? You can certainly report the ones with low NFE. This is important for verifying them later on by others.

Couple of high-level points are unclear to me; so I am now going with a “below threshold” rating. I am open to increasing it depending on authors response, if satisfactory.

---

> ### Author Response · Authors · 2023-11-19
>
> **W1.1: While I agree that figuring out good FID numbers for existing benchmarks is pretty good, it is not usually preffered to lock the contribution to very specific models/dataset/solver. If a new dataset/model/solver comes out in future, one need to perform the entire search again.**
>
> **A**: Thanks for raising this valuable concern. For new models, our experimental results in Tab.21 show that the searched schedules of S3 have the potential to be transferred across models.
>
> As for new tasks, we would like to highlight the efficiency of our search method, S3 (please refer to the analysis of search cost in the global response). The efficiency of S3 makes it feasible to employ S3 for new tasks.
>
> Specifically, on low-resolution datasets like CIFAR-10, S3 is about 100 $\times$ more efficient than training-based methods like Consistency Models [1]. And on high-resolution datasets like LSUN-Bedroom, S3 is around 700 $\times$ more efficient. Compared with a concurrent work of efficient diffusion sampler, DPM-Solver-v3 [2], the search cost of S3 is 4 $\times$ smaller and can achieve superior performance in the meantime on CIFAR-10.
>
> Moreover, our experiments find that the search cost of S3 can be easily decreased by evaluating fewer schedules with fewer images without significant performance dropping, according to the results in Tab.3 and Tab.16.
>
> [1] Consistency Model, Song et al., ICML 2023
>
> [2] DPM-Solver-v3: Improved Diffusion ODE Solver with Empirical Model Statistics, Zheng et al., NeurIPS 2023
>
> **W1.2: The major weakness I see is that the paper can be called “just a search” without any solid conclusion.**
>
> **A**: We conduct detailed analysis based on the search results and give some knowledge that can offer guidance for designing solver schedules in the future.
>
> We first **investigate the contributions of each component to the final results**. Specifically, we fix other components and search only for one component. We run the evolutionary search for a few steps and report the results. Below are the search results on CIFAR-10 and LSUN-Bedroom ("LOE" stands for "low order estimation" of derivatives). More details can be found in App C in the revision.
>
> |  |  | CIFAR10 |  |  | Bedroom |  |
> | --- | --- | --- | --- | --- | --- | --- |
> | NFE | 5 | 7 | 10 | 5 | 7 | 10 |
> | Baseline | 23.44 | 6.47 | 3.90 | 33.92 | 25.94 | 23.65 |
> | Timestep | **10.06** | **5.36** | **2.88** | **27.23** | **22.99** | **18.64** |
> | Order | 17.76 | 5.86 | 3.90 | 31.89 | 25.41 | 22.38 |
> | Prediction | 23.44 | 6.47 | 3.90 | 32.43 | 24.84 | 23.25 |
> | Corrector | 22.58 | 6.23 | 3.90 | 33.40 | 25.60 | 23.27 |
> | Scaling | 21.67 | 5.56 | 3.90 | 32.84 | 25.60 | 23.25 |
> | LOE | 23.44 | 5.91 | 3.84 | 33.92 | 25.62 | 23.52 |
>
> The results show that: (1) There is room for improvement across all components; (2) The timestep schedule has the largest influence on the performance; (3) The order schedule is the second most crucial component. According to these observations, we know that (1) Future work might get higher performance gains by focusing on refining the timesteps and order schedule. (2) When applying S3, if the search budget is tight, one can conduct the search on a search space only encompassing a subset of components.

---

> ### Author Response · Authors · 2023-11-19
>
> (Continued from the Previous Page)
>
> Then we **analyze the pattern of the searched schedules**.
>    - Timestep: Current solvers use "logSNR uniform" for low-resolution datasets and "time uniform" for high-resolution datasets. (1) We find that for low-resolution datasets, more time points should be placed when $t$ is small. (2) However, our observations reveal that the default "logSNR uniform" excessively emphasizes very small and large timesteps. According to our observation, we suggest putting more points at $0.2<t<0.5$ rather than putting too much at $t<0.05$. (3) For high-resolution datasets, we recommend a slightly smaller step size at $0.35<t<0.75$ on top of the uniform timestep schedule.
>   - Prediction Type: (1) We find that the data prediction model outperforms the noise prediction model by a large margin on low-resolution datasets, especially when the budget is very low (i.e., 4 or 5). (2) Current solvers commonly apply the data prediction model on all datasets.  But on large resolution datasets, we find noise prediction often outperforms data prediction, opposite to current methods. (3) When sampling with guidance in pixel space, noise prediction with dynamic thresholding can also outperform data prediction under very low NFE budgets. (4) Moreover, we find that noise prediction is not suitable when using a high Taylor order.
>   - Derivative estimation methods: Existing solvers use equal order for Taylor expansion and derivative estimation. USF proposes to decouple the Taylor order and derivative estimation order and apply a lower order for the latter. We find that a low-order derivative estimation is preferred for high-order Taylor expansion when the step size is very large. Oppositely, full-order estimation is more likely to outperform low-order estimation when the step size is not so large.
>    - Sample-wise Consistency: We find that the performance of solver strategies remains consistent among data samples. We use a very small number of images (1k in our main experiments, 500 and 250 in our ablation studies) to evaluate the solver schedule during the search, while using the standard setting for the final evaluation (i.e., 10k for ImageNet-256 and MS-COCO, 50k for other datasets). The performance ranking of solver strategies is highly consistent between evaluating with a few images and many images. Therefore, we can directly apply the solver strategies discovered with a few images to more samples.
>
> We update Sec 5.3 of the revision to discuss these observations.

---

> ### Author Response · Authors · 2023-11-19
>
> **W2: The USF isn’t really something new. It is just a “dumb” aggregation of a bunch of design options. It is only based on empirical evidence — but I wonder if it can be theoretically justified.**
>
> **A**: We would like to share our perspectives on the contribution of USF.
>   - Our work indeed draws inspiration from existing diffusion sampler methods. However, these methods are presented in very different ways, even when they are working on the same solver component and proposing similar strategies. The USF framework is the first to comprehensively summarize all components of diffusion samplers. It can unify all existing exponential integral-based solvers like DEIS (in the $\lambda$ domain), DPM-Solver, DPM-Solver++, and UniPC. In our view, a unified framework can provide guidance for future implementation and research.
>   - For instance, a unified framework can make the extension and combination of existing strategies much more straightforward. In fact, thanks to the USF framework, our work managed to propose and implement quite a few new strategies, like low-order derivative estimation, more types of scaling magnitude, and searchable time steps. In addition, USF also offers flexibility in incorporating other strategies like arbitrary prediction types (See the results in Tab.19 for a simple attempt at interpolation between data and noise prediction), free choice of corrector, and using arbitrary off-the-shelf numerical derivative estimators.
>   - In addition to the newly proposed single-step strategies mentioned above, USF also advocate the utilization of various strategies at different timesteps.
>   - To the best of our knowledge: (1) No existing work has systematically discussed the relationship between existing samplers. (2) The new strategies we proposed have not been used. (3) The idea of using different solver strategies at different timesteps is not fully explored either.
>
> For theoretical justification, we provide a discussion of the accuracy order of solution at App I to ensure the convergence of USF. It is worth mentioning that the performance of a solver is not only decided by accuracy order but also influenced by the coefficient of the high order error term $\mathcal{O}(h^{p})$. However, this coefficient changes with the trajectory value and function evaluation value, making it difficult to provide theoretical analysis other than the convergence order. This fact is an important motivation for us to propose searching optimal solver schedules.
>
> We're not sure we understand the concern on "theoretical justification" correctly. Besides convergence order analysis, what types of theoretical justification do the reviewer think are helpful for this work? We'll appreciate further suggestions on this point.
>
> **Q1, Q2 & Q3: it is not clear to me which part of the paper is being claimed to be the primary contribution — the search algorithm S3 itself, or the optimal solver schedules and corresponding FID numbers ? Is the paper supposed to be about the “search algorithm S3” or an empirical AutoML report ?**
>
> **A**: Our primary contribution lies both in the framework USF and the search method S3. We have discussed the novelty and contribution of USF in the above response. And here, we'd like to expand on the contribution of the S3 search method.
>
> As we've discussed in our paper, the idea of multi-stage predictor-based search is not new in the AutoML literature, e.g., [1]. What we want to claim as our contribution is that we propose an actionable and relatively efficient search method. The effectiveness of S3 has been validated by our extensive experiments, showing the possibility of sampling with very few NFEs without retraining the diffusion U-Net. And the search cost of S3 is moderate as we have analyzed in the public comment and App F.5 in the revision. This means S3 can be practically used for solver optimization for new tasks.
>
> Another on-the-side contribution is that our discovered schedules can be used directly, and empirical knowledge based on the search results can help guide future solver design.
>
> According to your concern, we have adjusted the summarization of contributions in the revision as follows.
> 1. (Primary) Design of the USF, which unifies existing solver methods and provides flexible extension space.
> 2. (Primary) Practical search method for solver schedule optimization, whose effectiveness is validated by extensive experiments.
> 3. The knowledge analysis based on the search results and the searched solvers that can be directly used.
>
> [1] GATES: A Generic Graph-based Neural Architecture Encoding Scheme for Predictor-based NAS, Ning et al., ECCV 2020

---

> ### Author Response · Authors · 2023-11-19
>
> **Q4: What exactly the paper “concludes” at the end the day — did you find any “pattern” in the optimal solver schedules that may motivate further work? I don’t see the exact configuration of the optimal solver schedules (found by S3) — are they written anywhere? You can certainly report the ones with low NFE. This is important for verifying them later on by others.**
>
> **A**: Thanks for your reminder! We have already demonstrate about some of our searched solver schedules in Sec G.5. More general patterns have been discussed in our response to W1.2. Additionally, we upload our code and these schedules to https://anonymous.4open.science/r/USF-anonymous-code-56E6 for reproduction. Due to the time limitation, we didn't have enough time to clean our code and systematically arrange more searched schedules. We will rearrange and open source our code together with all searched solver schedules in the future.

---

> ### Author Response · Authors · 2023-11-21
> **Follow-up**
>
> Dear reviewer qhfh,
>
> Thanks once again for your valuable time and helpful suggestions. Did we address the concerns satisfactorily? If you have any further comments, please do not hesitate to tell us. We are more than willing to provide further clarifications.

---

> > ### Comment · Reviewer_qhfh · 2023-11-22
> > **Thanks for the rebuttal**
> >
> > The authors clarified most of my doubts. They updated the paper in a meaningful way incorporating my and other's comments. They provided with a lot of insights into the method and what it found out. They also provided some of the searched schedules outright, which are important for someone to verify the claims.
> >
> > I say again that I am not an AutoML expert in any way. But, the response convienced me that this paper does provide some value. Hence I will update my score to "above threshold".

---

> > > ### Author Response · Authors · 2023-11-23
> > >
> > > Thank you again for your valuable questions and suggestions, which are really helpful for our paper!

---

### Official Review · Reviewer_HfC4 · 2023-10-29

**Soundness:** 3 good
**Presentation:** 3 good
**Contribution:** 3 good
**Rating:** 6
**Confidence:** 4

**Summary:**

To further improve the sample quality with less than 10 NFE, this paper proposed a unified sampling framework (USF) to study the optional strategies for solver. Specifically, USF splits the solving process of one step into independent decisions of several components, which also reveal that an appropriate solver schedules will benefit the quality and efficiency of diffusion models. Moreover, for each component, $\mathcal{S} ^{3}$ is proposed to search for optimal solver schedules in each time step automatically. Based on this framework, USF enables to achieve significant performance on various benchmark datasets when implemented to various diffusion models.

**Strengths:**

1.USF improves the sample quality with less than 10 NFE, which greatly benefits diffusion models in practical application. It unifies previous fast training-free samplers in every independent sampling process, which is very flexible and extensible.

2.The generated images are comparable to previous SOTA fast sampling method in various time step, which demonstrates the superior of USF.

3.In my humble opinion, the mathematical analysis is rigorous and can support the improvements on the experiment results.

**Weaknesses:**

1.The caption of Table 3 should be improved, it’s hard to figure our the meaning.

2.The search budget should be analysed more since it is very important to implement to previous diffusion models.

3.Why not demonstrated more categories of generated images when NFE is less? A fuller qualitative analysis would add strength on the USF.

4.How about the reproducibility of USF? Can you share the code? The open source code will help researchers continue to improve slow sampling speed since USF has great performance.

**Questions:**

The same as Weaknesses.

---

> ### Author Response · Authors · 2023-11-19
>
> **W1: The caption of Table 3 should be improved, it’s hard to figure our the meaning.**
>
> **A**: Tab.3 shows the FID results on the MS-COCO dataset. "Ours" stands for our method with default setting. "Ours-500" and "Ours-250" stand for generating only 500 and 250 images when evaluating solver schedules, which are demonstrated to show the performance of our method with a tighter search cost.
>
> Thanks for your questions. We have improved this caption in our revision for better understanding.
>
> **W2：The search budget should be analysed more since it is very important to implement to previous diffusion models.**
>
> **A**: Thank you for raising the need for more discussion on this valuable question. We discuss the search budget thoroughly in the public comment above. Please refer to it.
>
> **W3: Why not demonstrated more categories of generated images when NFE is less? A fuller qualitative analysis would add strength on the USF.**
>
> **A**: Thank you for your suggestions! We add more qualitative results in App J of our revision. Please refer to it.
>
> **W4: How about the reproducibility of USF? Can you share the code? The open source code will help researchers continue to improve slow sampling speed since USF has great performance.**
>
> **A**: Thank you for your advice! Due to the time limitation, we can only upload the unclean code and some of our searched solver schedules to https://anonymous.4open.science/r/USF-anonymous-code-56E6. We will surely open source our code and upload all searched solver schedules in the future after carefully rearranging them.

---

> ### Author Response · Authors · 2023-11-21
> **Follow-up**
>
> Dear Reviewer HfC4,
>
> Thanks once again for your valuable time and helpful suggestions. Did we address the concerns satisfactorily? If you have any further comments, please do not hesitate to tell us. We are more than willing to provide further clarifications.

---

### Official Review · Reviewer_7SgP · 2023-10-30

**Soundness:** 3 good
**Presentation:** 3 good
**Contribution:** 2 fair
**Rating:** 6
**Confidence:** 4

**Summary:**

In this paper, the authors proposed a predictor-based search method that optimizes the solver schedule to get a better time-quality trade-off of sampling. It is based on a unified sampling framework (USF) to study the optimal strategies for solver. Therefore, it enables us to take different solving strategies at different time steps, and according to the authors' study, it has a considerable potential to improve the sample quality and model efficiency. Finally, the authors used extensive experiments to validate the method of a large number of datasets such as CelebA, Cifar-10, LSUN, ImageNet and so on. Also, the authors applied S^3 to Stable-Diffusion models, and achieve 2x acceleration without losing the performance on text-to-image generation task on MS-COCO 256x256 dataset.

**Strengths:**

1. The reference list of this paper is very complete and the authors introduced their differences and connections quite clearly.
2. The biggest strength of this paper is that, the experiments are really solid for me.

**Weaknesses:**

1. The only weakness for me is on the writing style:
(a) In Equation (1), dt and dw_t should be \mathrm{d} t and \mathrm{d}w_t. Similarly, in other parts of this paper, the authors sometimes use dt and sometimes use \mathrm{d} t. The notation should be unified.
(b) In Equation (5), the \mathrm{d} t is missing in the right hand side of the equation.
(c) For the Table 1, I suggest you using \bigstrut before each line to make the text shown in the middle.

**Questions:**

1. Can you briefly tell me the differences between this paper and Lu et al's DPM-solver++, since both of them seem to use different strategies as multi-step solver? What kind of improvements have you done in your paper compared with DPM Solver++?
2. If time is available for you, maybe you can follow my suggestions on writing in the previous section.

---

> ### Author Response · Authors · 2023-11-19
>
> **W & Q2: Writing Style**
>
> **A**: We sincerely appreciate you for pointing out our typos. We have corrected them in the revision.
>
> **Q1: Can you briefly tell me the differences between this paper and Lu et al's DPM-solver++, since both of them seem to use different strategies as multi-step solver? What kind of improvements have you done in your paper compared with DPM Solver++?**
>
> **A**: The main contribution of DPM-Solver++ is the introduction of two new strategies: 1. multi-step solver and 2. data prediction model. Except for the first and last few steps that use a low-order solver, DPM-Solver++ uses the same solver strategies along timesteps.
>
> USF comprehensively summarizes tunable components of all diffusion solvers based on the exponential integral. The two new strategies of DPM-Solver++ are also incorporated in USF. See Tab.1 and App E.2 for details. Moreover, USF introduces different solver strategies among all timesteps, expanding the design space. With searched solver schedules, USF outperforms DPM-Solver++ by a large margin. In conclusion, DPM-Solver++ is a special case of USF, and USF is far more effective and flexible than DPM-Solver++.

---

> ### Author Response · Authors · 2023-11-21
> **Follow-up**
>
> Dear Reviewer 7SgP,
>
> Thanks once again for your valuable time and helpful suggestions. Did we address the concerns satisfactorily? If you have any further comments, please do not hesitate to tell us. We are more than willing to provide further clarifications.

---

> > ### Comment · Reviewer_7SgP · 2023-11-22
> >
> > Thanks for your explanation on the difference between you paper and DPM-Solver++. I also see your revisions on the writing style. Overall, I think I will still keep the "6" score.

---

> > > ### Author Response · Authors · 2023-11-23
> > >
> > > Thank you again for your valuable questions and suggestions, which are really helpful for our paper!

---

### Official Review · Reviewer_VK39 · 2023-10-31

**Soundness:** 3 good
**Presentation:** 3 good
**Contribution:** 3 good
**Rating:** 6
**Confidence:** 4

**Summary:**

In this paper, the authors propose a sampling framework for diffusion models for the systematic study of solving strategies for diffusion models. Specifically, the authors explore the solver schedule in the following aspects: timestep discretization, prediction type, starting point, solver order, derivative estimation, and corrector usage. Experiment results show that the proposed method boosts the performance, especially in the regime of very few NFEs.

**Strengths:**

$\cdot$ This paper combines the AutoML methods into the diffusion sampling procedure. The authors explore the search space over timestep discretization, prediction type, starting point, solver order, derivative estimation, and corrector usage.

$\cdot$ The experiment results are appealing, especially in a few NFEs regime. The proposed method overperforms all baseline methods.

$\cdot$ The paper is overall well-written and the structure is clear.

**Weaknesses:**

$\cdot$ The experiment part seems to lack exact data on the computational cost of the overall pipeline of proposed methods. The overall pipeline needs iterative sampling images under a given sampling configuration, evaluating the performance and training of the proposed predictor models.

$\cdot$ The content of section 4.2 is not well-organized and needs further explanation.

**Questions:**

Does the predictor model need to be trained separately on different NFEs?

---

> ### Author Response · Authors · 2023-11-19
>
> **W1:The experiment part seems to lack exact data on the computational cost of the overall pipeline of proposed methods. The overall pipeline needs iterative sampling images under a given sampling configuration, evaluating the performance and training of the proposed predictor models.**
>
> **A**: Thank you for pointing out our oversight of inadequate discussion of this question. We have discussed the search overhead thoroughly in the global response above. Please refer to it.
>
> **W2:The content of section 4.2 is not well-organized and needs further explanation.**
>
> **A**: Thank you for your question. Sec 4.2 describes the workflow of our predictor-based multi-stage search algorithm, S3. We first initialize the population (a set of truly evaluated solver schedules) with several baseline schedules under all NFE budgets. Then we evolutionary sample a bunch of solver schedules from the search space and truly evaluate them to expand the initial population. Then we iteratively conduct the following 3 steps: 1. Training predictor with all schedules in the population; 2. Using the predictor to sample a new bunch of promising solver schedules from the search space; 3. Evaluating these new schedules to expand the population. After all iterations end, we finally choose the schedule with the best performance in the population as our search results. This method works because of the efficiency of evolutionary search and the acceleration brought by the predictor for exploring the search space.
>
> We have reconstructed Sec 4.2 in the updated paper for better presentation.
>
> **Q:Does the predictor model need to be trained separately on different NFEs?**
>
> **A**: The predictor doesn't need to be trained on different NFEs.  The predictor is trained on the whole solver schedule population, which contains schedules under all NFE budgets. We pad all schedules to the same length for the predictor. The predictor is used to give a predicted score of any solver schedule with any NFE in the range.

---

> ### Author Response · Authors · 2023-11-21
> **Follow-up**
>
> Dear Reviewer VK39,
>
> Thanks once again for your valuable time and helpful suggestions. Did we address the concerns satisfactorily? If you have any further comments, please do not hesitate to tell us. We are more than willing to provide further clarifications.

---

> > ### Comment · Reviewer_VK39 · 2023-11-22
> >
> > Thank you for the detailed response. The authors' response addressed most of my issues.

---

> > > ### Author Response · Authors · 2023-11-23
> > >
> > > Thank you again for your valuable questions and suggestions, which are really helpful for our paper!

---

### Official Review · Reviewer_HHFa · 2023-10-31

**Soundness:** 3 good
**Presentation:** 3 good
**Contribution:** 3 good
**Rating:** 6
**Confidence:** 3

**Summary:**

This paper proposes a framework to unify the existing Diffusion Probalisitc Model (DPM) solvers such as DPM-Solver, DPM-Solver++, UniPC and DESI. Under the framework, these methods can be represented with different options on some strategies and hyperparameters.
This paper shows the phenomenon that different solver strategies at different timesteps may yield different performances. This paper is inspired to design a schedule between different solving strategies during the sampling based on the unified framework. In this paper, the predicator-based method is proposed to schedule the solvers, which outperforms the SOTA solvers on several conditional datasets.

**Strengths:**

1. USF: This paper unifies the existing DPM solvers under a framework with detailed derivations and analysis, which provides a new perspective to review the DPM solvers and improve their performance in real applications.
2. $S^3$: This paper proposes the $S^3$ method which uses light predictors to search the solver schedule which demonstrates good performance on unconditional datasets with limited NEF.
3. Empirical Results: This paper has plentiful empirical studies. It demonstrates the situation that there is no single best solver under different timesteps. It also shows good performance compared with baselines and ablation studies on the consumption of generated images.

**Weaknesses:**

1. This paper illustrates the phenomenon that the suitable strategies vary among timesteps but has no further explanation on why this phenomenon happens. The proposed method paper may lack motivation.
2. This paper may lack novelty and contributions in the sense that it does not provide a further understanding of why suitable solvers are different on different timesteps.
3. On empirical results, this paper may lack the results when the NFE is larger than 10. The gap between the baselines and the proposed method may decrease with larger NFE.

**Questions:**

1. Can you provide an intuitive explanation of the phenomenon observed in the paper?
2. Can you add the experiment results on larger NFEs? Since the search method also has the time consumption, can you provide some experiment results on it?
3. Can you give a brief explanation of the meaning and effect of multi-stage in Algorithm 2?
4. Since the USF can include the method DEIS in the domain $t$, why do you not include this method as a baseline in experiments?

Here are some typos in App. D. In analyzing the current solvers, the index of $\tilde{x}_{i-1}$ may be incorrect.

They should be $t_{i-1}$ in DPM-Solver-2, DPM-Solver++(2S) and DPM-Solver++(2M).

When using the Assumption in the Appendix, it may not be Thm A.1, A.2 etc.

---

> ### Author Response · Authors · 2023-11-19
>
> **W1, W2 & Q1: This paper may lack motivation, novelty and contributions in the sense that it does not provide a further understanding of why suitable solvers are different on different timesteps. Can you provide an intuitive explanation of the phenomenon observed in the paper?**
>
> **A**: Thank you for pointing out this question! Below we discuss the intuitions of the phenomenon "suitable strategies are different at different steps". We first lay out the high-level analyses, and then expand on each component.
>   - High level
>     - **The curvature of the ODE trajectory might vary across timesteps.** When $t$ is small, the neural network can reconstruct $x_0$ from $x_t$ more precisely and the ODE trajectory points to the target data. Intuitively, the curvature in the small $t$ region is small. Conversely, when $t$ is not close to 0, the ODE trajectory does not necessarily orient to the final target point and could have a large curvature [1]. Therefore, the curvature of the ODE trajectory might change over time. Some other papers have discussed similar findings. [2] demonstrates the pattern of VP-SDE in Sec.3, from which we can see that $x_t$ changes slowly at large $t$ and changes rapidly when $t$ is small; [1] finds that the trajectory orients to the mean value of the distribution at large $t$ while points to the data when $t$ is small. Different curvature leads to varying truncation errors, making certain solver strategies more suitable for specific timesteps.
>       - [1] Minimizing Trajectory Curvature of ODE-based Generative Models, Lee et al., ICML 2023
>       - [2] Elucidating the Design Space of Diffusion-Based Generative Models, Karras et al., NeurIPS 2022
>     - **The fitting error of the neural network also varies among timesteps.** The distributions of input for the neural network are different among timesteps. Therefore, the fitting tasks for the diffusion U-Net have unequal difficulty at different timesteps [3], leading to different distances between the predicted score and the ground truth score. Since the discretized solving error is related to the model's fitting error, it is natural for solver strategies to change across timesteps to adapt to different fitting accuracies.
>       - [3] OMS-DPM: Optimize the Model Schedule of Diffusion Probabilistic Models, Liu et al., ICML 2023
>     - **The requirements of solving accuracy are different at different timesteps.** At larger $t$, deviant values might be corrected to the right trajectories in subsequent solving processes. Conversely, for small $t$, a high solving error could directly cause a drop in sample quality. Therefore, different timesteps might need different solving strategies.
>   - Specific to components: Here we discuss the analyses that motivate using different strategies for each component.
>     - Step size: For low-resolution datasets, the processing of details is more important than large-resolution datasets. Therefore, the step size should be smaller near $t=0$ than near $t=T$ for low-resolution datasets. This pattern has been adopted empirically for low-resolution datasets in existing solvers like DDIM and DPM-Solver, which motivates us to use different step sizes and search for better schedules than empirical patterns.
>     - Order: Existing high-order solvers (e.g., DPM-Solver, DPM-Solver++, and UniPC) empirically use low orders for the last several steps when the NFE budget is tight, indicating that the suitable orders for different timesteps may not be the same. We are inspired by this setting to use changeable orders at all timesteps for better performance.
>     - Prediction Type: The output of the data prediction model changes rapidly when $t$ is large and changes slowly when $t$ is small since the data ratio of $x_t$ changes from 0 to 1 along timesteps. For the noise prediction model, the trend is the opposite. The change speed of model output has a close relationship with the stability of high-order derivatives. Therefore, the proper prediction types at different timesteps might be different.
>     - Corrector: The ODE corrector can be viewed as a special ODE predictor with the involvement of the function evaluation $f_\theta(x_t, t)$ on the target point $t$. Thus, the above analysis also holds for corrector. We find that different corrector patterns are used by concurrent work [1] empirically to boost its results.
>       - [1] DPM-Solver-v3: Improved Diffusion ODE Solver with Empirical Model Statistics, Zheng et al., NeurIPS 2023
>     - The choices of different components are coupled with each other. For example, low Taylor order and low derivative estimation order are intuitively suitable for a large step size; prediction type affects the choice of derivative scaling method. In addition, the choice of solver strategy at one step is impacted by the solution accuracy of previous steps. Therefore, we apply different strategies for all components at different timesteps and search for the optimal schedule.

---

> ### Author Response · Authors · 2023-11-19
>
> **W3 & Q2: This paper may lack the results when the NFE is larger than 10. The gap between the baselines and the proposed method may decrease with larger NFE.**
>
> **A**: Thanks for pointing out the need for larger NFE results. We report the FID result of baselines and our searched schedules below, where Baseline-B represents the baseline with the best setting and Baseline-W represents for the worst (see Sec 5 and Sec G.1 for details). We can see that our method still outperforms all baselines by a large margin at larger NFEs like 12, 15, and 20. Our method completely converges at NFE=15, much faster than existing solvers.
>
> |  |  | CIFAR-10 |  |  | CelebA |  |
> | --- | --- | --- | --- | --- | --- | --- |
> | NFE | 12 | 15 | 20 | 12 | 15 | 20 |
> | Baseline-W | 5.31 | 4.52 | 3.54 | 6.12 | 4.20 | 3.56 |
> | Baseline-B | 3.67 | 3.03 | 2.80 | 3.82 | 2.72 | 2.70 |
> | Ours | **2.65** | **2.41** | **2.41** | **2.32** | **2.06** | **2.06** |
>
> We would like to emphasize that when the NFE budget is adequate, the negative impact of sub-optimal empirical strategies diminishes. This fact can be verified by the decreasing gap between Baseline-W and Baseline-B. So, it is very reasonable that the gap between our method and baselines decreases. Considering this fact, we choose a more challenging and meaningful scenario to optimize, i.e., sampling under very few NFE budgets (4-10).
>
> **Q4: Since the USF can include the method DEIS in the domain $t$, why do you not include this method as a baseline in experiments?**
>
> **A**: For a more comprehensive comparison, we report the results of DEIS directly from their paper [3] and the results of our method below. We don't choose DEIS as a baseline in our paper because UniPC and DPM-Solver++ have already compared DEIS with their methods in their papers. DEIS fails to outperform DPM-Solver++ or UniPC in most cases.
>   - [3] FAST SAMPLING OF DIFFUSION MODELS WITH EXPONENTIAL INTEGRATOR, Zhang & Chen, ICLR 2023
>
> |  |  | CIFAR-10 |  |  |  | CelebA  |  |  |
> | --- | --- | --- | --- | --- | --- | --- | --- | --- |
> | NFE | 5 | 7 | 10 | 15 | 5 | 7 | 10 | 20 |
> | DEIS | 15.37 | - | 4.17 | 3.37 | 25.07 | - | 6.95 | 3.41 |
> | Ours | **7.65** | **3.91** | **3.09** | **2.41** | **5.17** | **3.80** | **2.73** | **2.06** |
>
> **Q3: Can you give a brief explanation of the meaning and effect of multi-stage in Algorithm 2?**
>
> **A**: Our main idea of S3 is to train a predictor to efficiently predict the performance of a solver schedule rather than evaluate its true performance by sampling a large number of images.
>
> The **meaning** of "multi-stage": we iteratively conduct the following process with each iteration called a "stage": 1. Training predictor with all truly evaluated solver schedules in the population. 2. Using predictor to explore within the search space. 3. Truly evaluating the promising solver schedules selected by the predictor to expand the population. In Alg.2, the $for$ loop starting from line 3 represents the index of stage. The overall workflow is demonstrated in Fig.4.
>
> The **effect** of multi-stage search: At the early stage of the search, the predictor is only required to explore in a limited sub-space. Therefore, only a small amount of schedule-performance data pair is needed to train the predictor, and the predictor can be quickly put into use. As the search process advances, the exploration range of the predictor increases. Thus, it needed to be updated with more data to ensure the accuracy of its prediction. Compared to the single-stage search (i.e., evaluate a large number of solver schedules in one go to train the predictor and perform the predictor-based search only once), the muli-stage method makes the sampling of solver schedule for true evaluation more effective since the predictor involves in this process.
>
> Thanks for your question! For a better presentation of the search method, we have reconstructed Sec 4.2 in the revision.
>
> **Q: Typos**
>
> **A**: We appreciate you very much for pointing out these typos in our paper. We have corrected these mistakes in our revision.

---

> ### Author Response · Authors · 2023-11-21
> **Follow-up**
>
> Dear Reviewer HHFa,
>
> Thanks once again for your valuable time and helpful suggestions. Did we address the concerns satisfactorily? If you have any further comments, please do not hesitate to tell us. We are more than willing to provide further clarifications.

---

> > ### Comment · Reviewer_HHFa · 2023-11-22
> >
> > Thank you for your detailed response. This addressed most of my concerns. I raised my score.

---

> > > ### Author Response · Authors · 2023-11-23
> > >
> > > Thank you again for your valuable questions and suggestions, which are really helpful for our paper!

---

### Author Response · Authors · 2023-11-19
**Discussion of the Search Cost**

We discuss the problem of the **search cost** here, which is a common concern by most reviewers. Below, we first **conduct a brief analysis of the computational cost** of our search method to demonstrate the efficiency of our search method, and then **compare the cost with other methods**.  Finally we **experimentally show that our method can still work well with much lower costs**. A more complete analysis can be found in App F.5 of the revision.

**Analysis of the search cost.** The total cost is mainly on generating images with sampled solver schedules. This overhead is equal to the time cost of N$\times$M$\times$S times neural inference of the diffusion U-Net, where N is the number of evaluated solver schedules, M is the number of images generated to calculate the metric, and S is the mean NFE of all solver schedules. We then test the runtime for diffusion U-Net to forward one batch of images on one single GPU and estimate the overall GPU time cost. We list the results below.

| Dataset | Total GPU hours | Device | Resolution |
| --- | --- | --- | --- |
| CIFAR-10 | 12.15 | NVIDIA 3090 | 32 |
| CIFAR-10(Ours-250) | 3.04 | NVIDIA 3090 | 32 |
| CelebA | 11.39  | NVIDIA 3090 | 64 |
| ImageNet-64 | 17.96 | NVIDIA 3090 | 64 |
| ImageNet-128 | 37.45 | NVIDIA A100 | 128 |
| ImageNet-256 | 130.33 | NVIDIA A100 | 256  |
| Bedroom | 68.36 | NVIDIA A100 | 256  |
| MS-COCO | 106.64 | NVIDIA A100 | 512 |
| MS-COCO(Ours-250) | 26.66 | NVIDIA A100 | 512 |


**Comparison with other methods.** We Compare our cost with two types of works aimming at fast sampling.
 - Training-based methods. Consistency model [1] is a popular and effective work which achieves very impressive performance with only 1-4 sampling steps. However, the training cost of this work is heavy. According to Tab.3 of its paper, consistency distillation/training on CIFAR-10 needs 800k$\times$512 samples, which is nearly 15 times more than 4000$\times$1000$\times$7 in our methods. Moreover, one iteration of training consistency models needs more than one forward pass (3 for CD and 2 for CT) and an additional backward process. Therefore, our search process is 90-100 times faster than training consistency models. For large resolution datasets, this acceleration ratio is even larger, e.g., 700 times faster on LSUN-Bedroom. Besides, our method does not have GPU memory limitations, while training-based methods suffer from this problem (e.g., consistency models are trained on 8 GPUs on CIFAR10 and 64 GPUs on LSUN)
     - [1] Consistency Model, Song et al., ICML 2023
 - Concurrent works of training-free sampler. DPM-Solver-v3 [2] is a concurrent work of improved diffusion sampler. Similar to USF, DPM-Solver-v3 needs to pre-compute three types of additional parameters before sampling. According to App.D of this paper, DPM-Solver-v3 costs 7$\times$8 GPU hours on CIFAR-10 and 11$\times$8 GPU hours on Stable Diffusion Models. We list the GPU hours needed for DPM-Solver-v3 and our method in the table below. We can see that the costs of these two methods on MS-COCO are close, while on CIFAR-10, our method is 4$\times$ faster. When using fewer images for metric calculation (Ours-250), we can be more efficient. It is worth mentioning that DPM-Solver-v3 mainly improves the prediction type components and can be incorporated in USF.
    - [2] DPM-Solver-v3: Improved Diffusion ODE Solver with Empirical Model Statistics, Zheng et al., NeurIPS 2023
| Dataset | CIFAR-10 | MS-COCO |
| --- | --- | --- |
| DPM-Solver-v3 | 56 | 88 |
| Ours | 12.15 | 106.64 |
| Ours-250 | 3.04 | 26.66 |

**Experiments with lower search cost.** We have given some experimental results in Sec 5.2 to show the influence of decreasing the search cost by generating fewer images per solver schedule. We then conduct more systematic experiments on CIFAR-10 to investigate the impact of both fewer images and less sampled schedules. Some of our results are demonstrated below. More detailed results can be found in App G.3 in the revision of our paper. The numbers in brackets are our acceleration ratio compared to default settings (consume 12.15 GPU hours). From these results, we can see that USF can still outperform baseline methods with a very limited search cost budget (e.g., 16x acceleration in the last row), showing the efficiency of S3.
| Schedule Number | Image Number | Total Acceleration Ratio |  |  | NFE |  |  |
| --- | --- | --- | --- | --- | --- | --- | --- |
| - | - |  | 4 | 5 | 7 | 9 | 10 |
| 2000(2x) | 1000(1x) | 2x (cost 6.07 GPU hours) | 14.94 | **6.86** | **4.17** | **3.13** | **2.67**  |
| 2000(2x)  | 500(2x) | 4x (cost 3.04 GPU hours) | **11.50** | 6.94 | 4.40 | 3.20 | 2.90 |
| 1000(4x) | 250(4x) | 16x (cost 0.76 GPU hours) | 15.42 | 8.11 | 4.84 | 3.46 | 3.33 |
|  | Baseline-B |  | 57.52 | 23.44 | 6.47 | 4.30 | 3.90 |

---

### Author Response · Authors · 2023-11-19

We thank all the reviewers for the insightful questions and suggestions on our work! We appreciate the recognition of the contribution of USF (HHFa, HfC4) and S3 (HHFa, VK39, qhfh), extensive and promising results (all reviewers), and the presentation of our work (VK39, 7SgP, Hfc4, qhfh). We're also thankful for all the concerns and suggestions. All the concerns and suggestions are helpful and inspiring. We will provide answers to all the questions. Please let us know if there are further questions.

According to the suggestions, we have made the following adjustments to our paper. We mark all updated content in blue in our revision.
  - Major
    - Revise Sec 4.2 for a better presentation of the multi-stage method.
    - Add a more detailed discussion and comparison of our search cost in App F.5.
    - Add analysis on the potential of each component in App C.
    - Add more experimental results, including the results with lower search cost in App G.3, results under larger NFE budgets in App G.2.
  - Minor
    - Adjust the contributions in Sec 1.
    - Adjust the insights in Sec 5.3
    - Improve the caption of Tab 3.
    - Add a description of some searched solver schedules in App G.6.
    - Add more generated images as qualitative results in App J.
    - Fix typos.

---

### Meta-Review · Area_Chair_qTZk · 2023-12-07

**Metareview:**

This paper introduces a Unified Sampling Framework (USF) to consolidate existing Diffusion Probabilistic Model (DPM) solvers. The authors further propose a search strategy to determine the optimal combination of solvers, considering factors such as timestep discretization, prediction type, starting point, solver order, derivative estimation, and corrector usage. The primary goal is to enhance sample quality with fewer than 10 function evaluations (NFE). Extensive experiments demonstrate state-of-the-art performance on various datasets. The authors, in their revision, include discussions and comparisons on search complexity and offer high-level insights into using different samplers at various stages. The reviewers express overall satisfaction with the paper's current state and unanimously recommend acceptance. Congratulations on the excellent work!

**Justification For Why Not Higher Score:**

The technical novelty of the paper is somewhat limited, as there is no more principled conclusion or investigation on which properties of the problem favor certain samplers and components of the sampler. The paper could be further improved with some discussion or theoretical analysis on the trade-off or the optimal choice of samplers in different tasks.

**Justification For Why Not Lower Score:**

The experiments in this paper are very solid. Based on them, the authors demonstrated the necessity of using different samplers at different stages of diffusion models and also presented an effective strategy to work around this issue.

---

### Decision · Program_Chairs · 2024-01-16

Accept (poster)